# The evaluating study of the momentum and heat exchange process of two surface layer schemes during the severe haze pollution in Jing-Jin-Ji in east China

Yue Peng[1,2], Hong Wang[1,2], Yubin Li[3], Changwei Liu[3], Tianliang Zhao[2], Xiaoye Zhang[1], Zhiqiu Gao[3,4], Tong Jiang[5], Huizheng Che[1], Meng Zhang[6]

[1] State Key Laboratory of Severe Weather/Institute of Atmospheric Composition, Chinese Academy of Meteorological Sciences (CMAS), Beijing 100081, China

[2] Collaborative Innovation Center on Forecast and Evaluation of Meteorological Disasters/Key Laboratory for Aerosol-Cloud-Precipitation of China Meteorological Administration, Nanjing University of Information Science and Technology, Nanjing 210044, China

[3] Key Laboratory of Meteorological Disaster of Ministry of Education/Collaborative Innovation Center on Forecast and Evaluation of Meteorological Disasters, School of Remote Sensing and Geomatics Engineering, Nanjing University of Information Science and Technology, Nanjing 210044, China

[4] State Key Laboratory of Atmospheric Boundary Layer Physics and Atmospheric Chemistry, Institute of Atmospheric Physics, Chinese Academy of Sciences, Beijing 100029, China

[5] National Climate Center, China Meteorological Administration, Beijing 100081, China

[6] Beijing Meteorological Service, Beijing 100089, China

*Correspondence to:* Hong Wang (wangh@cma.gov.cn)

**Abstract.** The turbulent flux parameterization schemes in surface layer are crucial for air pollution modeling. The pollutants prediction by atmosphere chemical model exist obvious deficiencies, which may be closely related to the uncertainties of the momentum and sensible heat fluxes calculated in the surface layer. The differences of two surface layer schemes (the Li and MM5 scheme) were discussed and the performance of the two schemes was evaluated based on the observed momentum and sensible heat fluxes in Jing-Jin-Ji in east China. The results showed that the aerodynamic roughness length $z_{0m}$ and the thermal roughness length $z_{0h}$ play an important role in the flux calculation. Compared with the Li scheme, ignoring the difference between the two in the MM5 scheme induced great error in the calculation of sensible heat flux (e.g., the error was 54% at Gucheng station). Besides the roughness lengths, the algorithms of universal functions as well as the roughness sublayer also resulted in certain errors in the MM5 scheme. In addition, the magnitudes of $z_{0m}$ and $z_{0h}$ have significant influence on the two schemes. The large $z_{0m}$ and $z_{0m}/z_{0h}$ in megacity with rough surface (e.g., Beijing) resulted in much larger differences of momentum and sensible heat fluxes by Li and MM5, compared with the small $z_{0m}$ and $z_{0m}/z_{0h}$ in suburban area with smooth surface (e.g., Gucheng). The Li scheme better characterized the evolution of atmospheric stratification than the MM5 scheme in general, especially for the transition stage from unstable to stable atmospheric stratification corresponding to the PM$_{2.5}$ accumulation. The bias of momentum and sensible heat fluxes from Li were lower about 38% and 43% respectively than those from MM5 during this stage. This study indicates the superiority of the Li scheme in the describing of the regional atmospheric stratification, and also suggests the improving possibility of severe haze

prediction in Jing-Jin-Ji in east China by coupling it into the atmosphere chemical model online.
**Key words**: surface layer; turbulent flux parameterization; roughness length; numerical modeling; air pollution
**1 Introduction**
Adequate air quality modeling relies on accurate simulations of meteorological conditions, especially in planetary
boundary layer (PBL) (Hu et al., 2010; Cheng et al., 2012; Xie et al., 2012). The PBL is closely coupled to the earth's surface
by turbulent exchange processes. As the bottom layer of PBL, the surface layer (SL) reflects the surface state by calculating
momentum, heat, water vapor and other fluxes, and influences the atmospheric structure by turbulent transport process.
Many studies have illustrated the important roles of meteorological factors in the SL in the formation of air pollution. They
demonstrated that weak wind speed, high relative humidity (RH) and strong temperature inversion are favorable for the haze
concentrating (Zhang et al., 2014; Yang et al., 2015; Liu et al., 2017; Zhong et al., 2017). The strong stable stratification and
weak turbulent are mainly responsible for many haze events. The relationship between flux and atmospheric profile in the
atmospheric surface layer is a key factor for air pollution diffusion, especially under stable stratification conditions (Li et al.,
2017). However, the study of stable boundary layer still has some uncertainties due to the poor description of surface
turbulent motion. The simulating study on a severe haze in east China by the Weather Research and Forecasting/Chemistry
(WRF-Chem) model concluded that there is lower ability of current PBL schemes in distinguishing the diffusion between
haze days under stable condition and clean days under unstable condition (Li et al., 2016a). Another study (Vautard et al.
2012) on mesoscale meteorological models also pointed out a systematic overestimation of near-surface wind speed in a
stable boundary layer and its possible contribution to the underestimation of the $PM_{2.5}$ pollution. In addition,
atmospheric conditions in both the PBL and upper layers are strongly dependent on the turbulent fluxes which are computed
in the SL (Ban et al., 2010). Flux parameterization in the SL plays an important role in studies of the hydrological cycle and
weather prediction (Yang et al., 2001; Li, 2014). An adequate SL scheme is crucial to provide an accurate atmospheric
evolution by numerical models (Jiménez et al., 2012) and hence it may introduce important impacts on air pollution
simulation.
The bulk aerodynamic formulation based on Monin-Obukhov similarity theory (hereinafter MOST, Monin and
Obukhov, 1954) is usually employed to calculate surface fluxes in numerical models. Turbulent fluxes are parameterized by
wind, temperature, humidity in the lowest layer in model and temperature and humidity in surface. Many international
scholars verified the MOST using of field experiments and then proposed the universal functions, the commonly used of
which is Businger-Dyer (BD) equation (Businger, 1966; Dyer, 1967). With the development of observation technology, the
coefficients in the BD equation have been further modified (Paulson, 1970; Webb, 1970; Businger et al., 1971; Dyer, 1974;
Högström, 1996). In addition to the BD equation, some other schemes have been put forward and they performed better

especially for the strongly stable stratification (Holtslag and De Bruin, 1988; Beljaars and Holtslag, 1991; Cheng and Brutsaert, 2005). The schemes can be divided into two types according to the computing characteristics. One type is called as iterative algorithm (Paulson, 1970; Businger et al., 1971; Dyer, 1974; Högström, 1996; Beljaars and Holtslag, 1991), and it keeps the MOST completely with less approximation so that the results can be more precise. However, it needs to take much more steps to converge and hence the CPU time is consuming which reduces the computational efficiency of modeling (Louis, 1979; Li et al., 2014); The other one is called as non-iterative algorithm (Louis et al., 1982; Launiainen, 1995; Wang et al., 2002; Wouters et al., 2012). There is no need for loop iteration in the calculation due to the approximate treatment. This algorithm is much simpler and less CPU time-consuming, but the results are based on the loss of the calculation accuracy.

A new non-iterative scheme proposed by Li et al. (2014; 2015, Li hereinafter) speeds up effectively under a higher accuracy compared with some classic iterative computation. It is remarkable that this new scheme just have been theoretically evaluated and it has never been applied in any models. Haze pollution occurs frequently in recent years in east China. The concentration of $PM_{2.5}$ may reach up to 1000 $\mu g \cdot m^{-3}$ in the Beijing-Tianjin-Hebei (Jing-Jin-Ji) region in winter (Wang et al., 2014) while it was generally underestiamted by current air quality models (Zhang et al., 2015; Li et al., 2016a; Liu et al., 2017). The Li and another classic SL scheme (Zhang and Anthes, 1982, MM5 hereinafter) are compared in details in this study. The observed momentum and sensible heat flux data covering once complete haze process at Gucheng station was used to evalute the two schemes focsuing on the transition stage from unstable to stable atmospheric stratification corresponding to the $PM_{2.5}$ accumulation. The evaluation is in the view of both local and regional scales. This offline study may provide the prerequisite for the online coupling the Li scheme into atmosphere chemical model in the future.

**2 Theory**

The definition of the momentum and sensible heat flux as well as the detailed algorithms of the Li and MM5 schemes are introduced in this section.

**2.1 Introduction of the momentum and sensible heat flux**

The turbulent fluxes from ground surface are defined as follows:

$$\tau = \rho u_*^2, \tag{1a}$$

$$H = -\rho c_p u_* \theta_*. \tag{1b}$$

Where $\tau$ is the momentum flux, $H$ is the sensible heat flux, $\rho$ is the air density, $c_p$ is the specific heat capacity at constant pressure. $u_*$ and $\theta_*$ are the friction velocity and the temperature scale respectively, and they represent the intensity of the vertical turbulent flux transport and they are approximately independent on height in the SL.

Both the Li and MM5 schemes are calculated with bulk flux parameterization. As an important dimensionless parameter

related to the stability, the bulk Richardson number $Ri_B$ is defined as

$$Ri_B = \frac{gz(\theta-\theta_g)}{\theta u^2}. \tag{2}$$

Where g is the acceleration of gravity, $z$ is the reference height which is the lowest level in the model, $\theta$ is the mean
potential temperature at height z, $\theta_g$ is the surface radiometric potential temperature, $u$ is the mean wind speed at height z.
Thus, $Ri_B$ can be computed through meteorological variables at least two levels.
**2.2 The Li scheme**
This new scheme employ non-iterative algorithm to compute the surface fluxes. Its basic idea is to parameterize the
stability parameter $\zeta$ directly with $Ri_B$ and roughness lengths ($z_{0m}$ and $z_{0h}$). Specifically, bulk transfer coefficients of the
momentum and sensible heat fluxes ($C_M$ and $C_H$) are expressed as

$$C_M = \frac{u_*^2}{u^2} = \frac{\tau}{\rho u^2}, \tag{3a}$$

$$C_H = \frac{u_*\theta_*}{u(\theta-\theta_g)} = -\frac{H}{\rho c_p u(\theta-\theta_g)}. \tag{3b}$$

Based on MOST and considering the roughness sublayer (RSL) effect at the same time, the relationship between the
bulk transfer coefficients and the profile functions corresponding to wind and potential temperature are usually expressed as

$$C_M = \frac{k^2}{\left[\ln\frac{z}{z_{0m}}-\psi_M\left(\frac{z}{L}\right)+\psi_M\left(\frac{z_{0m}}{L}\right)+\psi_M^*\left(\frac{z}{L}, \frac{z}{z_*}\right)\right]^2}, \tag{4a}$$

$$C_H = \frac{k^2}{R\left[\ln\frac{z}{z_{0m}}-\psi_M\left(\frac{z}{L}\right)+\psi_M\left(\frac{z_{0m}}{L}\right)+\psi_M^*\left(\frac{z}{L}, \frac{z}{z_*}\right)\right]\left[\ln\frac{z}{z_{0h}}-\psi_H\left(\frac{z}{L}\right)+\psi_H\left(\frac{z_{0h}}{L}\right)+\psi_H^*\left(\frac{z}{L}, \frac{z}{z_*}\right)\right]}. \tag{4b}$$

Where $k$ is the von Kármán constant which is 0.4 in both two schemes, $R$ is the Prandtl number which is 1.0 in the
two schemes, $z_{0m}$ and $z_{0h}$ are the aerodynamic roughness length and the thermal roughness length, respectively. $\psi_M$ and
$\psi_H$ are the integrated stability functions for momentum and sensible heat, respectively, which are also called universe
functions. $L$ is the Obukhov length ($\zeta = \frac{z}{L}$), $\psi_M^*$ and $\psi_H^*$ are the correction functions accounting for RSL effect, $z_*$ is the
RSL height. It is clear to see that the calculation of the momentum and sensible heat fluxes requires $C_M$ and $C_H$ (or $u_*$ and
$\theta_*$), and there are 3 key points to get them:
1. $z_{0m}$ and $z_{0h}$. $z_{0m}$ and $z_{0h}$ are two key parameters in the bulk transfer equations. Their definitions and influence
will be discussed in Sect. 4.1. Note that both $z_{0m}$ and $z_{0h}$ are taken into account by the Li scheme. In other words,
the Li scheme distinguishes these two important surface parameters effectively as they generate from different
mechanisms.
2. $\zeta$. The determination of $\zeta$ is the most crucial problem for the Li scheme. In fact, this new scheme includes two
parts. The first part was proposed for atmospheric stable stratification condition (Li et al., 2014), and the second part
then extended the scheme to unstable condition (Li et al., 2015). For stable condition, the calculation procedure for a
given group of $Ri_B$, $z_{0m}$ and $z_{0h}$ is the following: (1) find the region according to $z_{0m}$ and $z_{0h}$; (2) find the section

according to the region and $Ri_B$ with Eq. (5) and given coefficients; (3) calculate $\zeta$ using Eq. (6) and given coefficients.

$$Ri_{Bcp} = \sum C_{mn}(\log L_{0M})^m (L_{0H} - L_{0M})^n, \quad (5)$$

$$\zeta = Ri_B \sum C_{ijk} Ri_B^i L_{0M}^j (L_{0H} - L_{0M})^k. \quad (6)$$

Where $C_{mn}$ and $C_{ijk}$ are the coefficients in Tables in Li et al. (2014). $L_{0M} = \ln \frac{z}{z_{0m}}$, $L_{0H} = \ln \frac{z}{z_{0h}}$. $m, n = 0, 1, 2$, and $m + n \leq 3$; $i, j, k = 0, 1, 2, 3$, and $i + j + k \leq 4$. Similarly, for unstable condition, eight regions are divided according to the method from Li et al. (2015). For each of the regions, $\zeta$ is carried out by following:

$$\zeta = Ri_B \frac{L_{0M}^2}{L_{0H}} \sum C_{ijk} \left(\frac{-Ri_B}{1-Ri_B}\right)^i L_{0M}^{-j} L_{0H}^{-k}. \quad (7)$$

Where $C_{ijk}$ is listed in Li et al. (2016b), and $i = 0, 1$; $j, k = 0, 1, 2, 3$; $i + j + k \leq 4$.

3. Universal function. It is also a key factor in flux calculation. The form of universal function here is adopted from Cheng and Brutsaert (2005) under the stable condition (Eqs. (8a), (8b)) and it is adopted from Paulson (1970) under the unstable condition (Eqs. (9a), (9b)):

$$\psi_M(\zeta) = -a \ln\left[\zeta + (1 + \zeta^b)^{\frac{1}{b}}\right], \quad \zeta > 0 \text{ (stable)}, \quad (8a)$$

$$\psi_H(\zeta) = -c \ln\left[\zeta + (1 + \zeta^d)^{\frac{1}{d}}\right], \quad \zeta > 0 \text{ (stable)}, \quad (8b)$$

$$\psi_M(\zeta) = 2 \ln\frac{1+x}{2} + \ln\frac{1+x^2}{2} - 2\arctan(x) + \frac{\pi}{2}, \quad \zeta < 0 \text{ (unstable)}, \quad (9a)$$

$$\psi_H(\zeta) = 2\ln\frac{1+y}{2}, \quad \zeta < 0 \text{ (unstable)}. \quad (9b)$$

Where $a = 6.1$, $b = 2.5$, $c = 5.3$, $d = 1.1$, $x = (1 - 16\zeta)^{1/4}$, $y = (1 - 16\zeta)^{1/2}$.

In addition, the RSL effect is taken into account in the Li scheme. The definitions and influence of RSL will also be discussed in Sect. 4.1. De Ridder (2010) proposed the expression of $\psi_M^*$ and $\psi_H^*$:

$$\psi_M^* \left(\zeta, \frac{z}{z_*}\right) = \phi_M \left[\left(1 + \frac{\upsilon}{\mu_M z/z_*}\right)\zeta\right] \frac{1}{\lambda}\ln\left(1 + \frac{\lambda}{\mu_M z/z_*}\right) e^{-\mu_M z/z_*}, \quad (10a)$$

$$\psi_H^* \left(\zeta, \frac{z}{z_*}\right) = \phi_H \left[\left(1 + \frac{\upsilon}{\mu_H z/z_*}\right)\zeta\right] \frac{1}{\lambda}\ln\left(1 + \frac{\lambda}{\mu_H z/z_*}\right) e^{-\mu_H z/z_*}. \quad (10b)$$

Where $\upsilon = 0.5$, $\mu_M = 2.59$, $\mu_H = 0.95$, $z_* = 16.7 z_{0m}$, $\lambda = 1.5$. $\phi_M$ and $\phi_H$ are universal functions before integration. Here, set $\chi_M = 1 + \frac{\upsilon}{\mu_M z/z_*}$, $\chi_H = 1 + \frac{\upsilon}{\mu_H z/z_*}$:

$$\phi_M(\chi_M \zeta) = 1 + a \frac{\chi_M\zeta + (\chi_M\zeta)^b[1+(\chi_M\zeta)^b]^{\frac{1-b}{b}}}{\chi_M\zeta + [1+(\chi_M\zeta)^b]^{\frac{1}{b}}}, \quad \zeta > 0 \text{ (stable)}, \quad (11a)$$

$$\phi_H(\chi_H \zeta) = 1 + c \frac{\chi_H\zeta + (\chi_H\zeta)^d[1+(\chi_H\zeta)^d]^{\frac{1-d}{d}}}{\chi_H\zeta + [1+(\chi_H\zeta)^d]^{\frac{1}{d}}}, \quad \zeta > 0 \text{ (stable)}, \quad (11b)$$

$$\phi_M(\chi_M \zeta) = (1 - 16\chi_M\zeta)^{-1/4}, \quad \zeta < 0 \text{ (unstable)}, \quad (12a)$$

$$\phi_H(\chi_H\zeta) = (1 - 16\chi_H\zeta)^{-1/2}, \quad \zeta < 0 \text{ (unstable).} \qquad (12b)$$

**2.3 The MM5 scheme**

The MM5 scheme is a classic one which is widely applied in modeling investigation (Hu et al., 2010; Wang et al., 2015a, b; Tymvios et al., 2017). This scheme dose not distinguish $z_{0h}$ from $z_{0m}$, thus the roughness length here is expressed as $z_0$. For unstable condition, the function forms are given by Eqs. (16a) and (16b) following Paulson (1970), and for stable condition, the atmospheric stratification conditions are subdivided into three cases according to Zhang and Anthes (1982) and the function forms are given by Eqs. (13), (14), and (15).

(1) Strongly stable condition ($Ri_B \geq 0.2$):

$$\psi_M = \psi_H = -10 \ln\frac{z}{z_0}. \qquad (13)$$

(2) Weakly stable condition ($0 < Ri_B < 0.2$):

$$\psi_M = \psi_H = -5 \left(\frac{Ri_B}{1.1-5Ri_B}\right) \ln\frac{z}{z_0}. \qquad (14)$$

(3) Neutral condition ($Ri_B = 0$):

$$\psi_M = \psi_H = 0. \qquad (15)$$

(4) Unstable condition ($Ri_B < 0$):

$$\psi_M = 2\ln\frac{1+x}{2} + \ln\frac{1+x^2}{2} - 2\arctan(x) + \frac{\pi}{2}, \qquad (16a)$$

$$\psi_H = 2\ln\frac{1+y}{2}, \qquad (16b)$$

where $x = (1 - 16\zeta)^{1/4}$, $y = (1 - 16\zeta)^{1/2}$.

This scheme calculates turbulent fluxes of the momentum and sensible heat with $u_*$ and $\theta_*$. In order to avoid the huge difference of $u_*$ through the two computation, $u_*$ is arithmetically averaged with its previous value with Eq. (17), and a lower limit of $u_* = 0.1$ m/s is imposed to prevent the heat flux from being zero under very stable conditions. According to the profile functions of wind and temperature near the ground, $\theta_*$ then is deduced by Eq. (18).

$$u_* = \frac{1}{2}\left(u_* + \frac{ku}{\ln\frac{z}{z_{0m}}-\psi_M}\right), \qquad (17)$$

$$\theta_* = \frac{k(\theta-\theta_g)}{R[\ln\frac{z}{z_{0h}}-\psi_H]}. \qquad (18)$$

The calculation procedure of the Li scheme is the following: (1) determine $Ri_B$、$z_{0m}$ and $z_{0h}$ according to the observation data; (2) calculate $\zeta$ with $Ri_B$、$z_{0m}$ and $z_{0h}$; (3) calculate the momentum and sensible heat fluxes under different conditions. The MM5 scheme is summarized as follows: (1) determine the universal functions according to the values of $Ri_B$ and $z_0$; (2) calculate the $u_*$ and $\theta_*$ with the meteorological variables and flux data; (3) derive the turbulent fluxes. Compared with other non-iterative schemes including MM5, the Li scheme can be applied to the full range of

roughness status $10 \leq \frac{z}{z_{0m}} \leq 10^5$ and $-0.5 \leq \ln \frac{z_{0m}}{z_{0h}} \leq 30$ under whole conditions $-5 \leq Ri_B \leq 2.5$. In addition, there are
three obvious differences between the Li and MM5 schemes: (1) Li distinguishes $z_{0h}$ from $z_{0m}$ but MM5 does not
distinguish them; (2) the two schemes apply different universal functions under stable condition; (3) Li considers the RSL
effect while MM5 ignores it.
**3 Observational data and methods**

The observational fluxes used in this study measured at Gucheng station from December 1, 2016 to January 9, 2017.

Gucheng station (115.40 °E, 39.08 °N) is located at Gucheng County, Baoding, Hebei province and it is about 110km
southwest of Beijing (Fig. 1a). This station has a farmland site where rice is planted in summer and wheat in winter. The
surroundings are mainly farmland and scattered villages (Fig. 1b). At Gucheng station, the momentum and sensible heat
fluxes near surface were measured by the eddy correlation flux measurement system. The system is mainly composed of a
sonic anemometer (CSAT3) and a gas analyzer (LI-7500). They are set up at 4m height above surface ground. The measured
fluxes are used to evaluate the two schemes as well as estimate the roughness lengths. The measured meteorological
variables including wind speed and direction, temperature, humidity, pressure, radiation are used to calculate the momentum
and sensible heat fluxes both in the Li and MM5 schemes. Note the observed meteorological data were from Gucheng station
and national basic automatic weather stations in Jing-Jin-Ji in east China, respectively. Hourly surface $PM_{2.5}$ mass
concentration in Baoding and Beijing from China National Environmental Monitoring Centre (http://www.cnemc.cn/) were
also used in this paper.
**3.1 Data processing**

To obtain accurate flux data, quality control has been performed for the observational data, including: (1) eliminate the

outliers and the data in rainy days; (2) double rotation and WPL correction (Webb et al., 1980); (3) omit the dataset when the
wind speed is less than 0.5m/s. In addition, the wind field especially the wind direction has a great impact on the value of
$z_{0m}$, so it is necessary to understand the situation at Gucheng station. Fig. 2 shows the distribution frequency of wind speed
and wind direction at Gucheng during the observation (December 1, 2016 ~ January 9, 2017). The wind speed is stable
during this period and the maximum is no more than 5 m/s and most of them are about 1 ~ 2 m/s. The wind direction is
relatively uniform except for the southeast wind (135°).
**3.2 Determination of surface skin temperature**

The surface skin temperature at Gucheng station is calculated from the radiation data by the following formula:

$$R_{lw}^{\uparrow} = (1 - \varepsilon_s)R_{lw}^{\downarrow} + \varepsilon_s \sigma T_g^4, \tag{19}$$

where $R_{lw}^\uparrow$ and $R_{lw}^\downarrow$ are the surface upward longwave radiation and long wave radiation incident on the surface,
respectively. $\sigma$ is the Stephen Boltzmann constant, $\sigma = 5.67 \times 10^{-8} \text{Wm}^{-2}\text{K}^{-4}$. $T_g$ is the surface skin temperature, $\varepsilon_s$ is
the surface emissivity which is the prerequisite for calculating $T_g$. Many researches estimated $\varepsilon_s$ and the range of the values
is always 0.9 ~ 1 (Stewart et al., 1994; Verhoef et al., 1997). According to the semi-empirical method in Yang et al. (2008),
$\varepsilon_s$ is estimated when the RMSE is minimal. In this paper, the Li and MM5 schemes were used to estimate the $\varepsilon_s$ value (as
shown in Fig. 3). It is clear that the $\varepsilon_s$ value corresponding to the minimum RMSE is not very sensitive to the choice of two
schemes. When $\varepsilon_s$ is 1, the RMSE has the minimum value. Thus, this experiment takes 1 as the optimal value of $\varepsilon_s$.
**3.3 Determination of roughness length $z_{0m}$ ($z_{0h}$)**
Using the observed momentum and sensible heat fluxes and the meteorological variables including wind speed,
temperature, humidity and pressure after quality control at Gucheng station, $z_{0m}$ and $z_{0h}$ were derived by Eqs. (20a) and
(20b) following Yang et al. (2003) and Sicart et al. (2014).
$$\frac{u_*}{u} = \frac{k}{\ln\frac{z}{z_{0m}} - \psi_M}, \tag{20a}$$

$$\frac{\theta_*}{(\theta - \theta_g)} = \frac{k}{R[\ln\frac{z}{z_{0h}} - \psi_H]}. \tag{20b}$$

During the observation period, the crops stopped growing and the height did not exceed 0.1 m, so the zero-plane
displacement height was ignored hence the reference height z was taken as 4m. The observation time was too short (about 1
month) to consider the effect of seasonal variations on roughness lengths. Thus, $z_{0m}$ and $z_{0h}$ were assumed as two fixed
values. Based on the variables and formulae mentioned above, the roughness lengths at Gucheng are derived: $z_{0m} =$
0.0419 m, $z_{0h} = 0.0042$ m.
**4 Results and discussion**
The RSL, roughness length and their influence on the calculation of turbulent flux are discussed in detail in this section.
The Li and MM5 schemes are offline tested and evaluated during the haze pollution from December 13 to 23, 2016.
**4.1 The influence of RSL and roughness length on the calculation of turbulent flux**
The RSL is usually defined as the region where the flow is influenced by the individual roughness elements as reflected
by the spatial inhomogeneity of the mean flow (Florens et al., 2013). In the RSL, turbulence is strongly affected by
individual roughness elements, and the standard MOST is no longer valid (Simpson et al., 1998). Therefore, it is necessary to
consider the RSL effect in the calculation of turbulent fluxes, especially for the rough terrain such as forest or large cities.
$z_{0m}$ is defined as the height at which the extrapolated wind speed following the similarity theory vanishes. It is mainly
determined by land-cover type and canopy height after excluding large obstructions. In models, $z_{0m}$ is always based on the
look-up table which is related to land-cover types. In this study, $z_{0m}$ was simply classified based on the research of Stull
(1988) listed in Table 1. It can be seen in Table 1 that the rougher underlying surface corresponds to the larger value of $z_{0m}$.
$z_{0h}$ is the height at which the extrapolated air temperature is identical to the surface skin temperature. Some early researches
assumed that $z_{0m}$ was equal to $z_{0h}$ (Louis, 1979; Louis et al., 1982). However, the assumption is not applicable in reality
because $z_{0m}$ and $z_{0h}$ have different physical meanings. Different treatment of $z_{0m}$ and $z_{0h}$ may introduce considerable
changes in the surface flux calculation (Launiainen, 1995; Kot and Song, 1998; Anurose and Subrahamanyam, 2013). Many
studies removed the assumption that $z_{0m}$ was equal to $z_{0h}$ and made the schemes more applicable in the situation that $z_{0m}$
was not equal to $z_{0h}$ or the ratio of $z_{0m}$ to $z_{0h}$ was much large (Wouters et al., 2012; Li et al., 2014; Li et al., 2015).
Some field experiments even indicated the ratio $z_{0m}/z_{0h}$ has a diurnal variation (Sun, 1999; Yang, 2003; Yang, 2008). In
this study, we make the common assumption that the ratio $z_{0m}/z_{0h}$ is a constant.
Considering the lowest level in mesoscale models is usually about 10m, $z = 10$ m is set as the reference height. The
range of $Ri_B$ is set according to Louis82 (Louis et al., 1982) in the following discussion. Firstly, the effects of different
land-cover types (different $z_{0m}$ values) and RSL on flux calculation were discussed. Set $z_{0m} = z_{0h}$, corresponding to four
cases: $z_{0m}= 1, 0.5, 0.05, 0.001$ m. These cases correspond to large cities, forests, agricultural fields and wide water surface,
respectively. Fig. 4 shows the relationship between $C_M(C_H)$ and $Ri_B$ for different $z_{0m}$ values and treatment of RSL. It can
be seen that both RSL and $z_{0m}$ have impacts on $C_M$ and $C_H$. Ignoring the RSL effect results in lager $C_M$ and $C_H$,
compared with the results of original scheme considering the RSL. The difference induced by RSL is obvious only under the
rough surface. For example, the difference under $z_{0m}= 1$ is obviously greater than other $z_{0m}$ settings, and when $z_{0m}$ is
reduced to 0.05 or less, the RSL has little effect. Furthermore, the RSL contributes more to sensible heat transfer than to
momentum transfer under the same setting of $z_{0m}$. The effects of different land-cover types on $C_M$ and $C_H$ are much more
significant compared with RSL. The rougher the surface is (corresponding to the larger $z_{0m}$ value), the larger the $C_M$ $(C_H)$
is. In addition, there is a corresponding relationship between $C_M(C_H)$ and stability. The more unstable the atmosphere is, the
larger difference the value of $C_M(C_H)$ is and vice versa. Once $Ri_B$ exceeds the critical value (generally 0.2~0.25), the
transfer coefficients decline sharply but still above 0.
Secondly, the effects of difference between $z_{0m}$ and $z_{0h}$ as well as RSL on flux calculation are discussed. The
relationship between $z_{0m}$ and $z_{0h}$ can be expressed as $kB^{-1} = \ln \frac{z_{0m}}{z_{0h}}$. Over the sea, $z_{0m}$ is comparable to $z_{0h}$; over the
uniform vegetation surface (grassland, farmland, woodland), $kB^{-1}$ is about 2 ($z_{0m}/z_{0h} \approx 10$) (Garratt and Hicks, 1973;
Garratt, 1978; Garratt and Francey, 1978), which coincides with our results in Gucheng ($z_{0m} = 0.0419$ m, $z_{0h} =$
0.0042 m); over the surface with bluff roughness elements, the $kB^{-1}$ value may be very large. For example, in some large
cities, $kB^{-1}$ is even up to 30 ($z_{0m}/z_{0h} \approx 10^{13}$) (Sugawara and Narita, 2009). Therefore, the ratio $z_{0m}/z_{0h}$ varies over a
wide range. Fig. 5 shows the relationship between $C_M(C_H)$ and $Ri_B$ for different treatment of $z_{0m}/z_{0h}$. Set $z_{0m} = 1$ as a
large city case, $z_{0h}$=1, 0.01, $10^{-4}$, $10^{-6}$ m, and the large differences derived from the different ratios are displayed in Fig. 5.
The similar RSL effect can be found compared with Fig. 4. The differences induced by RSL are more obvious than that in
Fig. 4. The different treatment of ratio $z_{0m}/z_{0h}$ has great impact on turbulent flux transfer, particularly for sensible heat
transfer. It seems evident that when $z_{0h}$ is not equal to $z_{0m}$ ($z_{0m}/z_{0h}$=100 ~ $10^6$), the calculated $C_H$ is much small
compared to the treatment that $z_{0h}$ is equal to $z_{0m}$ ($z_{0m}/z_{0h}$=1). In addition, $C_M(C_H)$ decreases with the increase of
stability, and they decrease much slower when $z_{0h}$ is not equal to $z_{0m}$.

**4.2 Comparison of momentum and sensible heat fluxes calculated by the two schemes**

Using the obtained roughness lengths and the observations, the momentum and sensible heat flux were calculated by the

Li and MM5 schemes. Firstly, $z_{0m}$ and $z_{0h}$ were set as 0.0419 and 0.0042 respectively in the Li scheme, $z_0$ was equal to
$z_{0m}$ in the MM5 scheme to calculate the momentum and sensible heat fluxes and the results are shown in Figs. 6a and 6b. It
can be seen that compared with MM5, Li performs better with higher regression coefficient and determination coefficient.
For the momentum fluxes, the regression coefficient by Li is 0.6795 and that by MM5 is 0.5598, indicating that the error of
Li is 12% lower than that of MM5. For sensible heat fluxes, the regression coefficient by Li is 0.7967 and that by MM5 is
1.7994. The latter is much larger than 1, that is, the MM5 scheme obviously overestimates the sensible heat due to it does not
distinguish $z_{0h}$ from $z_{0m}$. Then, make $z_0$ equal to 0.0042 in the MM5 scheme to re-calculate the sensible heat fluxes as
shown in Fig. 6c. It can be seen the result has a great improvement after modifying $z_0$ value and the regression coefficient
by MM5 is 0.7363, indicating that the error was reduced by 54% after considering the $z_{0h}$ effect. The result indicates that
$z_{0h}$ plays a key role in both the SL scheme and the sensible heat flux (Chen and Zhang, 2009; Chen et al., 2011). However,
the error caused by Li is still 6% lower than that by MM5. This illustrates that in addition to the effect of roughness lengths,
the algorithm of the Li scheme itself is more reasonable than that of MM5 scheme.
**4.3 The specific performance of the two schemes in the severe haze pollution**

There were two obvious pollution processes during this observation period and one occurred during December 13 to 23,

2016. Fig. 7 shows the variations of hourly observed $PM_{2.5}$ concentration as well as the momentum and sensible heat fluxes
calculated by the Li and MM5 schemes at Gucheng station in this process. For the research purpose significance, only the
daytime (from 8:00 a.m. to 20:00 p.m.) was taken into account. Note in MM5, $z_0$ was 0.0419 when calculate momentum
fluxes and it was 0.0042 when calculate sensible heat fluxes. As shown in Fig. 7, the calculated results of momentum and
sensible heat fluxes for the two schemes are generally consistent with the trend of the observations. Specifically, for the
momentum fluxes (Fig. 7a), the results of two schemes have little difference when the values of observed momentum fluxes
are large or at the peak. When the observed momentum fluxes are small, the Li scheme results are close to or less than the
observations, while the MM5 scheme results are always higher than observations because of the limit of $u_* = 0.1$ in this
scheme. For the sensible heat fluxes (Fig. 7b), MM5 results are always lower while Li results are closer to observations
especially when the observed values are small. Furthermore, according to the evolution of $PM_{2.5}$ concentration, this haze
event was then divided into three stages: the clear stage (stage 1: 13~14), the transition stage (stage 2: 16~18) and the
maintenance stage (stage 3: 21~22). As shown in Fig. 7, in the clear stage (stage 1), the atmospheric stratification is unstable,
$PM_{2.5}$ concentration is low and there is a strong flux transport in the SL, the corresponding observations of the momentum
and sensible heat fluxes are relatively high and they vary greatly. In the transition stage (stage 2), the atmosphere is changing
from unstable to stable corresponding to hazes formation, the momentum and sensible heat fluxes gradually decreases and
the daily variation also decreases. In the maintenance stage (stage 3), the atmospheric stratification is very stable, and flux
transport in the SL is weak, both the momentum and sensible heat fluxes are at a low level. It can be seen that the Li results
are generally closer to the observations compared with MM5 results in all three stages.

Fig. 8 shows the probability distribution functions (PDF) of the difference of momentum fluxes (Figs. 8a, 8c, 8e, 8g)

and sensible heat fluxes (Figs. 8b, 8d, 8f, 8h) calculated by using the Li and MM5 schemes in different stages at Gucheng
station. In the whole pollution process, for the momentum fluxes (Fig. 8a), the PDF of the difference by Li tends to cluster in
a narrower range centered by 0, and the probability within $\pm 0.005 N\ m^{-2}$ is 46.82%, while this value by MM5 falls to 23.02%.
For the sensible heat fluxes (Fig. 8b), the PDF of the difference by Li is also more concentrated around 0 than that by MM5.
The probabilities of bias by Li and MM5 within $\pm 2.5 W\ m^{-2}$ are 32.54% and 13.49%, respectively. In stage 1, for the
momentum fluxes (Fig. 8c), the probability of bias by Li within $\pm 0.005 N\ m^{-2}$ is 38.09%. The bias of MM5 mainly
concentrates larger than 0, and the probability within $\pm 0.005 N\ m^{-2}$ is 14.29%. For the sensible heat fluxes (Fig. 8d), the
probability of Li bias within $\pm 2.5 W\ m^{-2}$ is 38.09%, the same as momentum fluxes. The bias of MM5 mainly concentrates
less than 0, and the probability within $\pm 2.5 W\ m^{-2}$ is 9.52%. In stage 2, the differences between the two schemes are more
obvious. The momentum and sensible heat fluxes bias by Li is the most concentrated around 0 in all cases, while the
distribution of bias by MM5 is similar to that in stage 1. Specifically, for the momentum fluxes (Fig. 8e), the probabilities of
bias by Li and MM5 within $\pm 0.005 N\ m^{-2}$ are 56.25% and 25.00%. For the sensible heat fluxes (Fig. 8f), the probabilities of
bias by Li and MM5 within $\pm 2.5 W\ m^{-2}$ are 40.62% and 6.25%. In stage 3, the difference between two schemes is small. For
the momentum fluxes (Fig. 8g), the probabilities of bias by Li and MM5 within $\pm 0.005 N\ m^{-2}$ are 22.73% and 27.27%. For
the sensible heat fluxes (Fig. 8h), the probabilities of bias by Li and MM5 within $\pm 2.5 W\ m^{-2}$ are both 36.36%.

Mean bias (MB), normalized mean bias (NMB), normalized mean error (NME) and root mean square error (RMES) of

Li and MM5 were calculated to test the two schemes. Table 2 shows that the Li scheme generally estimates better than the
MM5 scheme. In the whole haze process, the Li scheme underestimates the momentum fluxes by 3.63% relative to the
observations, while the MM5 scheme overestimates by 34.03%. The Li and MM5 schemes underestimate the sensible heat

fluxes by 15.69% and 50.22%, respectively. In the three stages, the Li scheme performs much better than the MM5 scheme in the stage 1 and stage 2, especially in stage 2 when atmospheric stratification transforms from unstable to stable condition, the difference between the Li and MM5 schemes are particularly significant. The Li and MM5 schemes overestimate the momentum fluxes by 7.68% and 45.56%, respectively, while Li and MM5 underestimate the sensible heat fluxes by 33.84% and 76.88%. The error of Li is much less than that of MM5. Considering the importance of atmospheric stratification in the generation and accumulation of $PM_{2.5}$ in stage 2, the Li scheme is expected to show better performance in online simulation of $PM_{2.5}$ than MM5.

Based on the good behavior of the Li scheme in Gucheng, the same experiment was performed at Beijing station to discuss the effect of different land-cover types on flux calculation for two schemes. For Beijing station, the assumption $z_{0m} = 1m$, $z_{0m}/z_{0h} = 10^6$ was made to represent the surface condition of megacity due to a lack in situ measurements of surface turbulent flux. As shown in Fig. 9, the evolution of $PM_{2.5}$ concentration at Beijing station was also divided into three stages (stage 1: 13~15; stage 2: 17~19; stage 3: 20~21) just like Gucheng in the discussion. Compare to Fig. 7, there is a significant increase in the difference of momentum and sensible heat fluxes between Li and MM5 in Fig. 9. To be specific, the momentum transfer in Beijing is obviously larger than that in Gucheng due to the great increase of the urban aerodynamic roughness length ($z_{0m}$). In the meanwhile, the difference between Li and MM5 has a further expansion at Beijing station compared with Gucheng. The sensible heat transfer by the Li scheme has great difference between clear days and pollution days, which is, the sensible heat transfer changes acutely in the stage 1 while it changes smoothly in the stage 2 and stage 3. The sensible heat transfer by the MM5 scheme is significantly different compared with Li result due to MM5 ignored the $z_{0m}$ effect, and the small number of $z_{0h}$ keeps the sensible heat fluxes at a low level in all three stages.

To quantify the differences between the two schemes, a relative difference is defined in percentage:

$$\Delta V = \left| \frac{V_{Li} - V_{MM5}}{V_{MM5}} \right| \times 100\%, \tag{21}$$

where $V_{Li}$ and $V_{MM5}$ are the momentum (or sensible heat) flux calculated by the Li and MM5 schemes, respectively. We obtained the relative differences at the two stations in the three stages through the statistics. It is clearly that the largest relative difference at Gucheng station is in the stage 2 and the value at Beijing station is in the stage 1. The differences in Beijing are always larger than that in Gucheng for each three stages. Specifically, the relative difference of momentum flux in stage 1, stage 2 and stage 3 increases by 73%, 34% and 27%, respectively, and the results of sensible heat flux are289%, 52% and 68%, respectively.

We further tested the two schemes in whole Jing-Jin-Ji region. Fig. 10 shows the mean momentum and sensible heat fluxes calculated by Li and MM5 schemes and their difference in Jing-Jin-Ji during the pollution episode. The assumption $z_{0m} = 0.1m$, $z_{0m}/z_{0h} = 10^3$ were used to represent the average condition of the underlying surface of Jing-Jin-Ji region. As shown in Fig. 10, the momentum fluxes calculated by Li are less than that by MM5 in most stations; the sensible heat

fluxes calculated by Li are usually larger than that by MM5. The result is consistent with the experiment of Gucheng station,
which further indicates the importance of considering $z_{0m}$ and $z_{0h}$ at the same time.

## 5 Conclusions

Using the observed momentum and sensible heat fluxes, together with conventional meteorological data including
pressure, temperature, humidity and wind speed from December 1, 2016 to January 9, 2017, including a severe pollution
episode from December 13 to 23, 2016, the differences and the performance of the two surface schemes were discussed and
evaluated in this paper. The evolution process of atmospheric stratification from unstable to stable corresponding to $PM_{2.5}$
increasing was mainly discussed. The contributions of roughness lengths ($z_{0m}$ and $z_{0h}$) and other factors in the SL schemes
to the momentum and sensible heat flux calculation were also discussed in details. The results are summarized as follows:
1) $z_{0m}$ and $z_{0h}$ have important effects on turbulent flux calculation in the SL schemes. Different values of $z_{0m}$ and
$z_{0h}$ in the schemes could induce great changes in flux calculation, indicating that it is very necessary and important to
distinguish $z_{0h}$ from $z_{0m}$. Ignoring the difference between the two in the MM5 scheme led to large errors in calculation of
sensible heat fluxes and this error in Gucheng is 54%. Besides the roughness lengths, the algorithms of two schemes are also
one of important factors. In addition, ignoring the effect of the RSL in schemes may also results in certain bias of momentum
and sensible heat fluxes in megacity regions which represent the rough underlying surface.
2) The effect of $z_{0m}/z_{0h}$ on turbulent fluxes is closely related to the land-cover types ($z_{0m}$). A rough land-cover type
(large $z_{0m}$) should be accompanied by a large value of $z_{0m}/z_{0h}$. The differences of momentum and sensible heat fluxes
calculated by Li and MM5 were much bigger in Beijing than that in Gucheng. This suggests that the MM5 scheme probably
induces bigger error in megacities with rough surface (e.g., Beijing) than it in suburban area with smooth surface (e.g.,
Gucheng) due to the irrational algorithm of MM5 scheme itself and the ignoring difference between $z_{0m}$ and $z_{0h}$.
3) The Li scheme generally performed better than the MM5 scheme in the calculation of both the momentum flux and
the sensible heat flux compared with observations at Gucheng station. The Li scheme made a better description in
atmospheric stratification which is closely related to the haze pollution, compared with the MM5 scheme. This advantage
was the most prominent in the transition stage from unstable to stable atmospheric stratification corresponding to the $PM_{2.5}$
accumulation. In this stage, the momentum flux calculated by Li was overestimated by 7.68% and this overestimation by
MM5 was up to 45.56%; the sensible heat flux by Li was underestimated by 33.84% while this underestimation by MM5
was even up to 76.88%. In most Jing-Jin-Ji region, the momentum fluxes calculated by Li were less than that by MM5 and
the sensible heat fluxes by Li were larger than that by MM5, which was consistent with Gucheng.
The offline study of the two SL schemes in this paper showed the superiority of the Li scheme for surface flux
calculation corresponding to the $PM_{2.5}$ evolution during the haze episode in Jing-Jin-Ji in east China. The study results offer
the prerequisite and a possible way to improve PBL diffusion simulation and then PM$_{2.5}$ prediction, which will be achieved
in the follow-up work of online integrating of the Li scheme into the atmosphere chemical model.

**Acknowledgments**

The study was supported by the National Key Project (2016YFC0203306), the National (Key) Basic Research and
Development (973) Program of China (2014CB441201), the National Key Project (2016YFC0203304)

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

**Table 1.** Typical values of $z_{0m}$ corresponding to various land-cover types

| $z_{0m}$/m | Land-cover types |
|---|---|
| 5~50 | Mountain (above 100m) |
| 1~5 | The center of large cities, hills or mountain area |
| 0.1~1 | Forests, the center of large towns |
| 0.01~0.1 | Flat grasslands, agricultural fields |
| $10^{-4}$~$10^{-3}$ | The snow surface, wide water surface, flat deserts |
| $10^{-5}$ | The ice surface |

**Table 2.** Statistics between the Li and MM5 schemes calculated turbulent flux at Gucheng station.

| | | Li | | | | MM5 | | | |
|---|---|---|---|---|---|---|---|---|---|
| | | MB | NMB | NME | RMSE | MB | NMB | NME | RMSE |
| Whole | $\tau$ | -0.0006 | -3.63% | 54.29% | 0.0142 | 0.0058 | 34.03% | 63.59% | 0.0143 |
| process | H | -2.2723 | -15.69% | 52.73% | 10.9649 | -7.2735 | -50.22% | 69.68% | 12.7946 |
| Stage 1 | $\tau$ | 0.0021 | 9.98% | 55.90% | 0.0172 | 0.0091 | 43.45% | 66.66% | 0.0169 |
| | H | 1.1775 | 5.79% | 37.87% | 10.5734 | -7.1891 | -35.34% | 55.70% | 13.1324 |
| Stage 2 | $\tau$ | 0.0013 | 7.68% | 44.50% | 0.0111 | 0.0079 | 45.56% | 56.81% | 0.0121 |
| | H | -4.5752 | -33.84% | 50.28% | 9.3995 | -10.3924 | -76.88% | 81.40% | 13.2553 |
| Stage 3 | $\tau$ | -0.0024 | -13.25% | 59.13% | 0.0144 | 0.0030 | 16.72% | 56.34% | 0.0138 |
| | H | 1.2818 | 11.39% | 66.31% | 11.4778 | -1.7479 | -15.52% | 65.90% | 10.4219 |

$*$ $\tau$: momentum flux; H: sensible heat flux; MB: mean bias; NMB: normalized mean bias; NME: normalized mean error;
RMSE: root mean square error. The units of MB and RMSE: $\mu g \cdot m^{-3}$.

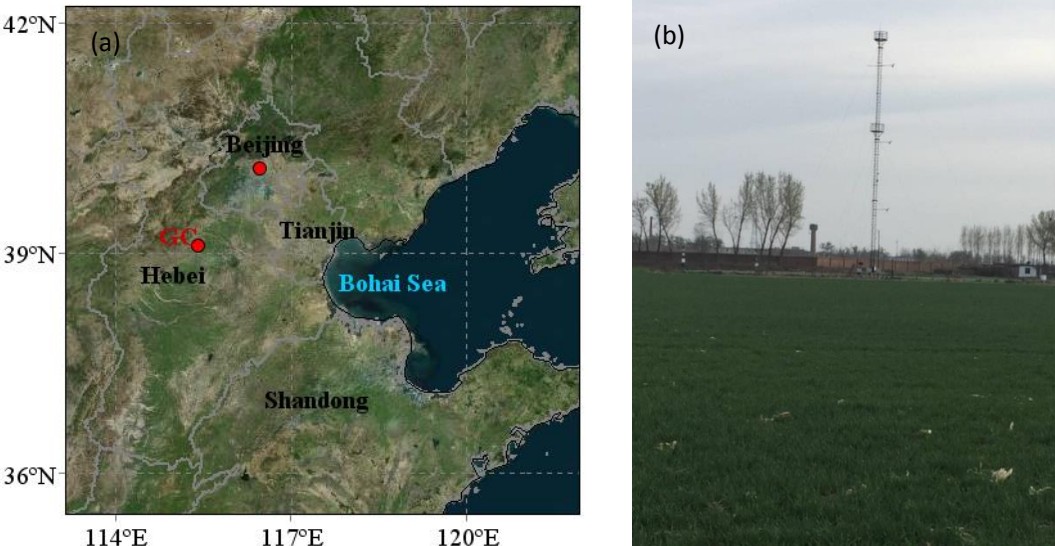


**Figure 1.** Location (a) and geographical environment (b) at Gucheng station. The map is from Bing Maps.


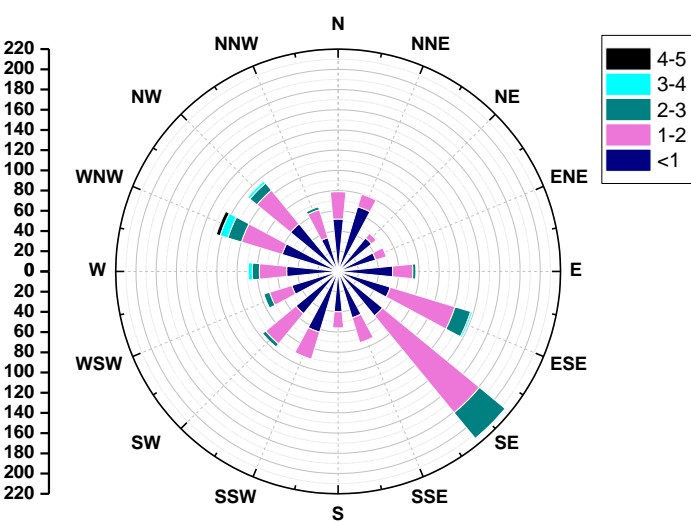


**Figure 2.** Wind Rose map at Gucheng station from December 1, 2016 to January 9, 2017.


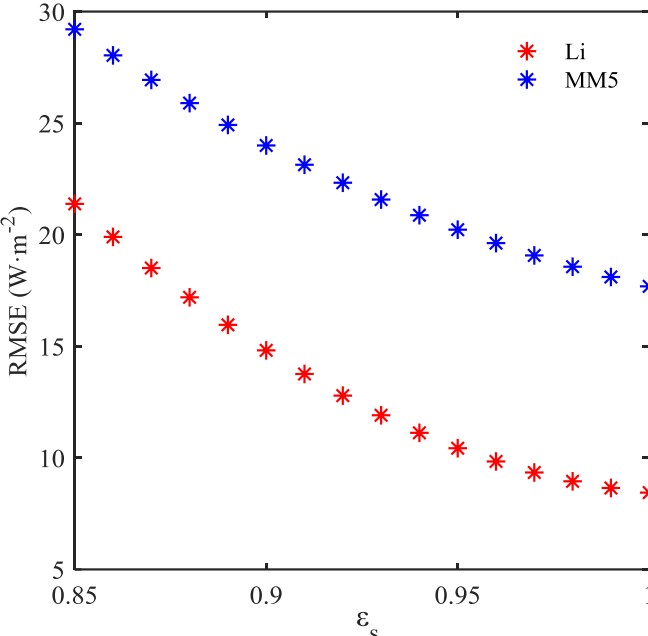


**Figure 3.** The surface emissivity $\varepsilon_s$ dependence of RMSE between observed near-neutral heat fluxes and parameterized

heat fluxes (red for Li and blue for MM5) at Gucheng station.




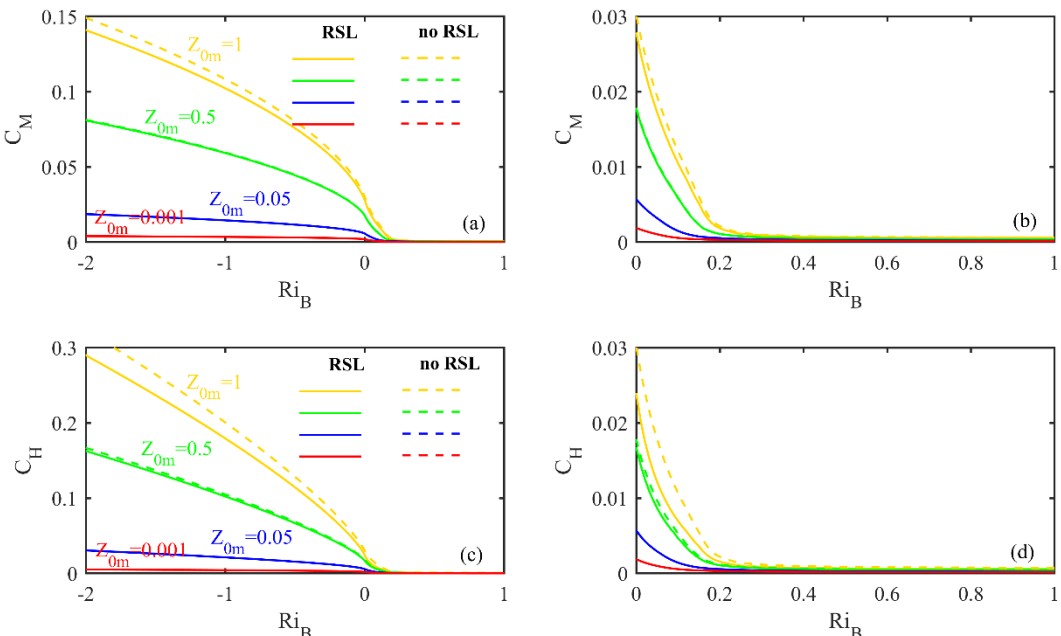


**Figure 4.** The relationship between $C_M(C_H)$ and $Ri_B$ for different $z_{0m}$ values and treatment of RSL. Solid lines:

considering the RSL effect; dotted lines: without the RSL effect.


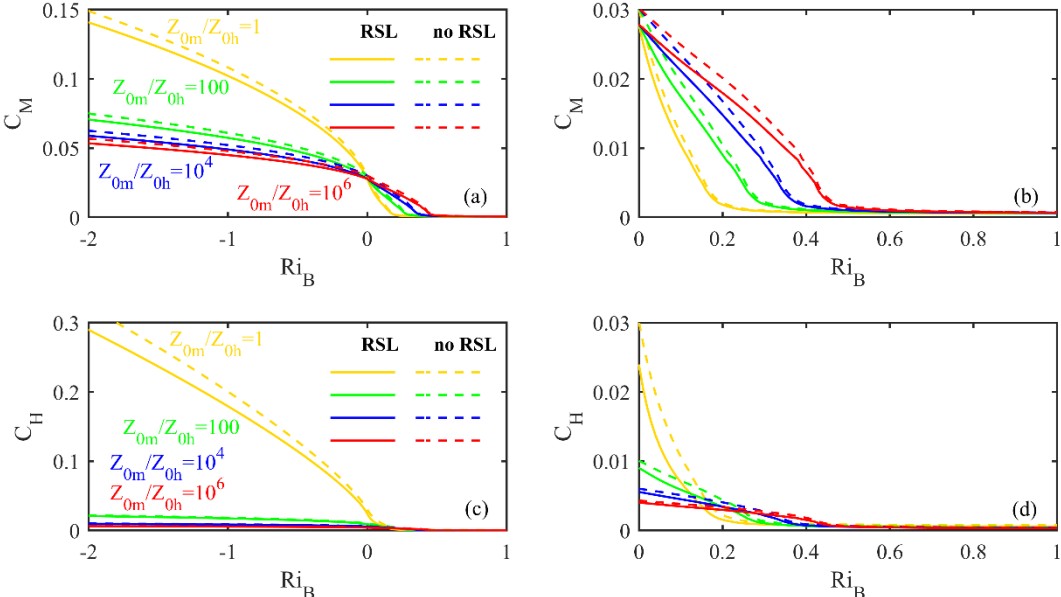

**Figure 5.** The relationship between $C_M(C_H)$ and $Ri_B$ for different ratios of $z_{0m}$ to $z_{0h}$ and treatment of RSL. Solid lines:
considering the RSL effect; dotted lines: without the RSL effect.

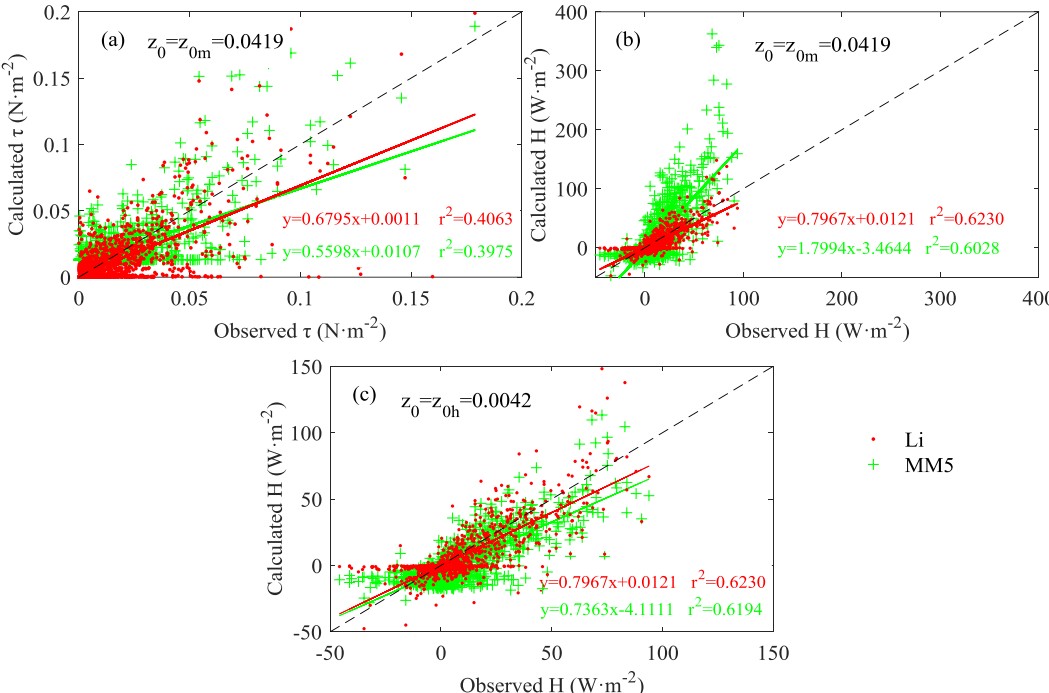


**Figure 6.** Comparison of calculated and observed fluxes at Gucheng station from December 1, 2016 to January 9, 2017. (a)
Momentum fluxes (MM5: $z_0 = 0.0419$); (b) sensible heat fluxes (MM5: $z_0 = 0.0419$); (c) sensible heat fluxes (MM5:
$z_0 = 0.0042$). Red dots: the Li scheme; green plus signs: the MM5 scheme.



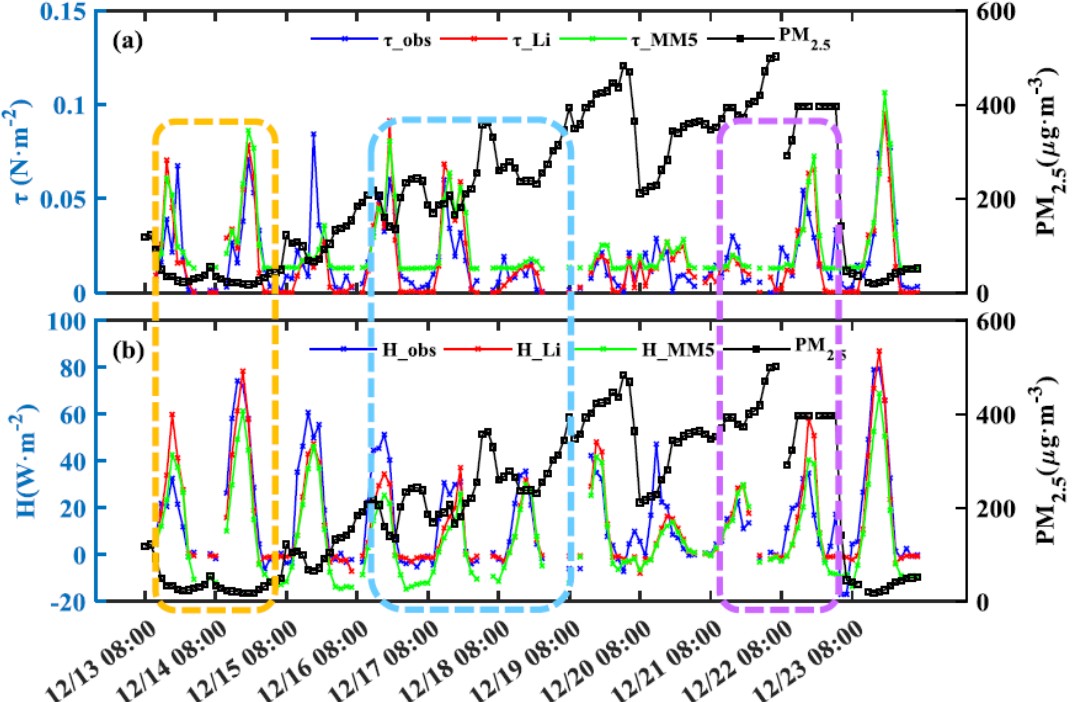

**Figure 7.** Variations of hourly turbulent fluxes and observed PM$_{2.5}$ at Gucheng station in daytime. (a) Momentum fluxes $\tau$
(blue line: observations; red line: the Li scheme; green line: the MM5 scheme) and PM$_{2.5}$ concentration (black line); (b)
sensible heat fluxes H (the same as $\tau$) and PM$_{2.5}$ concentration (black line). Yellow box: stage 1; blue box: stage 2; purple
box: stage 3.

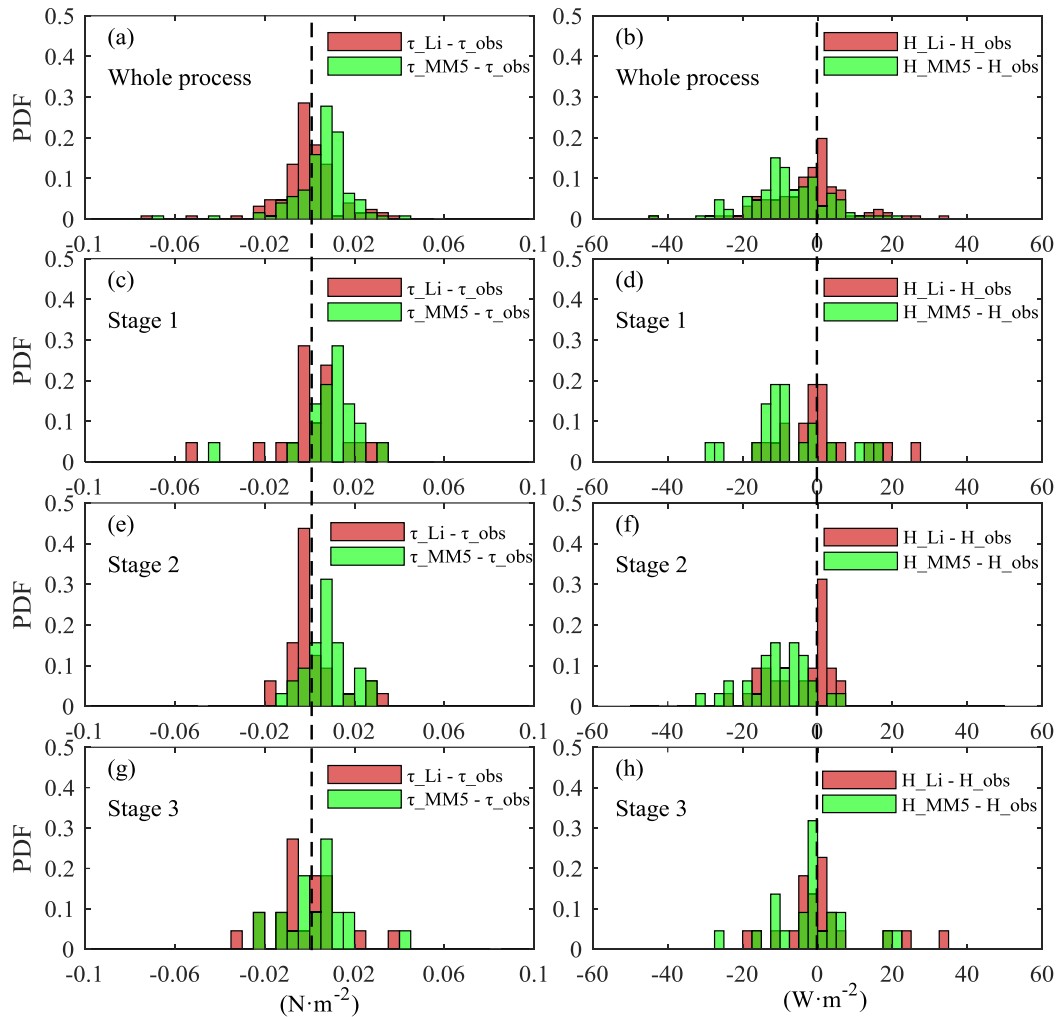

**Figure 8.** Probability distribution functions (PDF) of the difference between calculated fluxes (momentum fluxes: left; sensible heat fluxes: right) by using two schemes (the Li scheme: red bars; the MM5 scheme: green bars) and observations in different stages (a-b: whole process; c-d: stage 1; e-f: stage 2; g-h: stage 3).

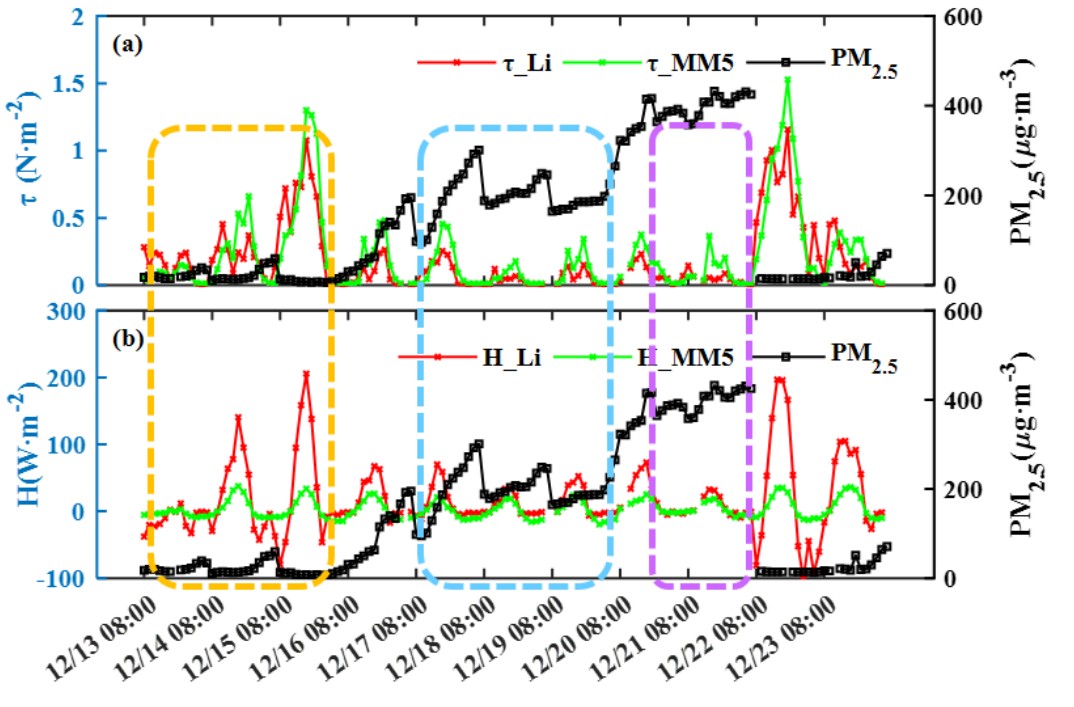

**Figure 9.** As in Fig. 7 but for Beijing station.

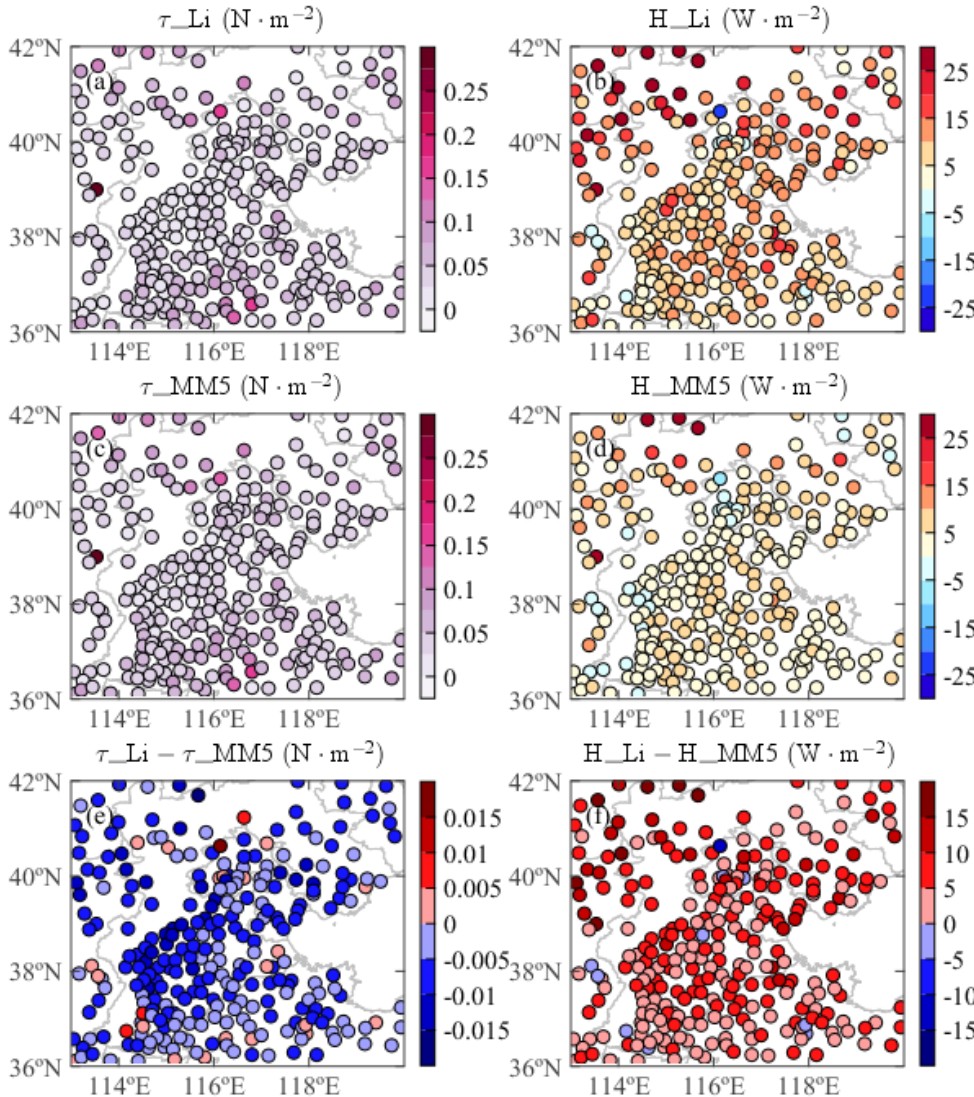

**Figure 10.** The mean momentum and sensible heat fluxes calculated by using two schemes (a-b: the Li scheme; c-d: the
MM5 scheme) and their difference (e: difference of the momentum fluxes; f: difference of the sensible heat fluxes) in
Jing-Jin-Ji during the haze episode (December 13 to 23, 2016).