# Peer review of "Evaluating the performance of two surface layer schemes for the"

_Atmospheric Chemistry and Physics, 2018_

## Referee Comment (RC1) · Anonymous Referee #3 · 22 May 2018

General comments

This study evaluated two surface layer schemes offline, and showed that the new Li scheme presents a better performance over the classic MM5 scheme in terms of the momentum and sensible heat fluxes. Given the importance of the surface exchange processes in a pollution episode and pollution forecast, an accurate representation of the surface processes would be required in a numerical model. This manuscript gave a rather good description about the two schemes, and the results did show that Li scheme may produce better agreement with observations especially in the transition stage of a haze episode. However, I have a few major concerns about this paper:

[Figure]
Interactive
comment

Major concerns:

1. What is the scientific contribution of this paper? The authors have well-addressed my comment in the quick report about the new improved surface layer scheme. However, as a scientific paper, I think the authors should also discuss and summarize the scientific findings of this study besides discussing the performance of the two schemes. For example,

1) How does the roughness length affect the turbulent fluxes and hence the pollution?
2) Does the roughness length plays a more important role in the transition stage of a pollution episode? And why?

2. There are a lot of grammar mistakes. Please carefully edit the manuscript to improve the language to ensure a better delivery of the scientific ideas and findings to the audience.

---

## Referee Comment (RC2) · Anonymous Referee #2 · 23 May 2018

This work evaluated the performance of a new surface layer scheme (Li) and a widely applied scheme (MM5) in simulating the momentum and sensible heat fluxes. Using the observational data in Gucheng station located in the southwest of Beijing from Dec 1, 2016, to Jan. 9, 2017, The authors found the Li scheme generally performed better than MM5 in calculating SL fluxes during the heavy pollution process. The study fits within the scope of the journal, and the manuscript is generally well written. The result presented is interesting as it shows the SL scheme performance in a polluted case. However, I found that some key details on the introductions are lacking and some of the discussions are not very well grounded.

Major comments:

1. The author should explicitly explain the scientific meaning of the paper. Since Li scheme has been published and evaluated in Li et al. (2014; 2015), why do we need additional evaluation using the observation during a severe haze episode from Gucheng station? I believe this evaluation may be necessary, but the authors need to illustrate clearly the specialty of this case. Also, the word "east China" appears several times in the paper. How did the author conclude Li generally performed better than MM5 in winter in east China since they only did one case in Beijing?

2. The role of surface layer (SL) scheme in air quality modeling needs to be further discussed in the introduction. The authors made sufficient introduction to the current status of SL. However, a detailed introduction of the importance of SL schemes in simulating pollution episode is somewhat lacking. In other words, the interactions between pollutant transportation, momentum and sensible heat (and how current SL schemes perform in momentum and sensible heat modeling) should be well established in the introduction part.

3. In the third conclusion (Line 342-343): The authors argued that "During the heavy pollution process, the calculated momentum and sensible heat fluxes by the Li scheme were better than those by the MM5 scheme generally". If the authors only compared simulated momentum and sensible heat to the observation, why this work emphasized the "heavily polluted conditions"? Future work may consider coupling SL scheme with atmospheric chemistry models to compare the modeled pollutant concentration with observation directly.

Minor comments:

1. Line 65-66: Why is the pollution episode important? The author may need to specify and add more discussion instead of arguing "few studies discussed it based on a pollution episode corresponding various atmospheric states."

[Figure]

2. Line 172-180: The observation and method should be introduced in further details. What is the spatial representativeness of the station? Can it represent the whole east China? If not, should add more cases in other parts of China or considering changing this word. What is the measuring height for the fluxes? (Could refer to Liu et al. 2016 as an example for the introduction)

3. Line 182-189: The data processing should be explained in further details and add more reference in data processing methods (Line 182-Line 189). For example, how was the quality control conducted? The reference for quality control may be included if they have been applied in the study (e.g., frequency response correction (Moore, 1986) and WPL correction (Webb et al., 1980), or quality control (Foken et al., 2004)).

4. Please explain why z = 10 m has been used (line 218)?

5. What variables have been used in Li and MM5 schemes? In the third part (Observational data and methods), the paper only introduced the data acquired from the Gucheng station, without specifying what variables would be used in the two schemes.

6. Straight from 5. Line 247, the authors mentioned: "Given the observational data, a dataset of Z0m (Z0h) then is generated". What variables were used in calculating Z0m and Z0h? This may be clarified in the third part (observational data and methods).

7. Line 250 to Lint 264: The author may consider comparing their conclusion with analysis from other papers (Chen et al. 2009; Chen et al. 2011). The reference used here is somewhat out of date.

8. In the Fig. 4, the authors showed the effect of the roughness length on flux calculation by choosing different z0m values. Since the z0m and Z0h has already been determined in the crop field, I feel it may not be necessary to discuss the influence of roughness length on the calculation of turbulent flux

9. Line 315-316: In the previous results and discussion, the authors only analyzed the superiority of Li scheme in modeling sensible heat and momentum flux. More analysis

is needed discussing the SL flux influence the air pollution process should be illustrated before concluding "the superiority of Li scheme in the air pollution modeling."

The reference listed here could be helpful:

Chen, Fei, and Ying Zhang. "On the coupling strength between the land surface and the atmosphere: From viewpoint of surface exchange coefficients." Geophysical Research Letters 36.10 (2009).

Chen, Yingying, et al. "Improving land surface temperature modeling for dry land of China." Journal of Geophysical Research: Atmospheres 116.D20 (2011).

Zheng, Donghai, et al. "Assessment of roughness length schemes implemented within the Noah land surface model for high-altitude regions." Journal of hydrometeorology 15.3 (2014): 921-937.

Liu, Ye, WeiDong Guo, and YaoMing Song. "Estimation of key surface parameters in semi-arid region and their impacts on improvement of surface fluxes simulation." Science China Earth Sciences 59.2 (2016): 307-319.

Moore C J. Frequency response corrections for eddy correlation systems[J]. Boundary-Layer Meteorology, 1986, 37(1-2): 17-35.

Webb E K, Pearman G I, Leuning R. Correction of flux measurements for density effects due to heat and water vapour transfer[J]. Quarterly Journal of the Royal Meteorological Society, 1980, 106(447): 85-100.

Foken T, Göockede M, Mauder M, et al. Post-field data quality control[M]//Handbook of micrometeorology. Springer, Dordrecht, 2004: 181-208.

---

## Author Comment (AC2) · 14 Aug 2018

Reply to Anonymous Referee #2:

We sincerely appreciate for the reviewer's careful dealing of our manuscript and valuable comments. We have read and discussed these comments in detail and answer them one by one in the followings. The corresponding revisions have also been added in the manuscript.

**General comments by Referee #2**

*This work evaluated the performance of a new surface layer scheme (Li) and a widely applied scheme (MM5) in simulating the momentum and sensible heat fluxes. Using the observational data in Gucheng station located in the southwest of Beijing from Dec 1, 2016, to Jan. 9, 2017, The authors found the Li scheme generally performed better than MM5 in calculating SL fluxes during the heavy pollution process. The study fits within the scope of the journal, and the manuscript is generally well written. The result presented is interesting as it shows the SL scheme performance in a polluted case. However, I found that some key details on the introductions are lacking and some of the discussions are not very well grounded.*

**Response:**

Thanks for the affirmation to our work. Yes, we agreed that some key points on the introduction were not enough and some discussions were not very well grounded. We have examined the introduction as well as whole text and the corresponding revisions have been added in the manuscript.

**Comment 1:** *The author should explicitly explain the scientific meaning of the paper. Since Li scheme has been published and evaluated in Li et al. (2014; 2015), why do we need additional evaluation using the observation during a severe haze episode from Gucheng station? I believe this evaluation may be necessary, but the authors need to illustrate clearly the specialty of this case. Also, the word "east China" appears several times in the paper. How did the author conclude Li generally performed better than MM5 in winter in east China since they only did one case in Beijing?*

**Response:**

The Li scheme consists of two parts (Li et al., 2014; 2015). The first part (Li et al., 2014) focused on the stable stratification, while the latter (Li et al., 2015) focused on the unstable conditions. The two parts have not been consolidated into a complete scheme in previous studies. In our study, the two parts were consolidated into one for both stable and unstable conditions. Furthermore, previous work (Li et al., 2014; 2015) was only compared with other iterative or non-iterative schemes. They have neither been compared with actual observations, nor evaluated under the transition process from unstable to stable conditions, which is essential and meaningful. We didn't introduce clearly in our old manuscript and we re-summarized this content in Line 74-83, Page 3 in the revised manuscript.

Yes, the word "east China" is not accurate in this paper. In fact, our study focuses on the Jing-Jin-Ji region in east China. We have replaced "east China" with "Jing-Jin-Ji" in the whole manuscript; In addition, we added Beijing station as well as Jing-Jin-Ji region to discuss the performance of Li and MM5 schemes for different land-cover types (added Figs. 9-10 and the related contents in the revised manuscript).

References:

1. Li, Y., Gao, Z., Li, D., Wang, L., and Wang, H.: An improved non-iterative surface layer flux scheme for atmospheric stable stratification conditions, Geosci. Model Dev., 7, 515-529, https://doi.org/10.5194/gmd-7-515-2014, 2014.

2. Li, Y., Gao, Z., Li, D., Chen, F., Yang, Y., and Sun, L.: An Update of Non-iterative Solutions for

Surface Fluxes Under Unstable Conditions, Bound.-lay. Meteorol., 156, 501-511, https://doi.org/10.1007/s10546-015-0032-x, 2015.

**Comment 2:** *The role of surface layer (SL) scheme in air quality modeling needs to be further discussed in the introduction. The authors made sufficient introduction to the current status of SL. However, a detailed introduction of the importance of SL schemes in simulating pollution episode is somewhat lacking. In other words, the interactions between pollutant transportation, momentum and sensible heat (and how current SL schemes perform in momentum and sensible heat modeling) should be well established in the introduction part.*

**Response:**

We agree that the introduction of the interactions between pollutant transportation, momentum and sensible heat was not enough and efficient, we read the new references list in the following and complemented the related contents in Line 42-52, Page 2 in the revised paper. The related references as follows were also added in the revised version.

References:

1. Zhang, R., Li, Q., and Zhang, R.: Meteorological conditions for the persistent severe fog and haze event over eastern China in January 2013, Sci. China Earth Sci., 57, 26–35, https://doi.org/10.1007/s11430-013-4774-3, 2014.

2. Yang, Y., Liu, X., Qu, Y., Wang, J., An, J., Zhang, Y., and Zhang, F.: Formation mechanism of continuous extreme haze episodes in the megacity Beijing, China, in January 2013, Atmos. Res., 155, 192–203, https://doi.org/10.1016/j.atmosres.2014.11.023, 2015.

3. Liu, T. T., Gong, S. L., He, J. J., Yu, M., Wang, Q. F., Li, H. R., Liu, W., Zhang, J., Li, L., Wang, X. G., Li, S. L., Lu, Y. L., Du, H. T., Wang, Y. Q., Zhou, C. H., Liu, H. L. and and Zhao, Q. C.: Attributions of meteorological and emission factors to the 2015 winter severe haze pollution episodes in China's Jing-Jin-Ji area, Atmos. Chem. Phys., 17, 2971–2980, https://doi.org/10.5194/acp-17-2971-2017, 2017.

4. Zhong, J., Zhang, X., Dong, Y., Wang, Y., Liu, C., Wang, J., Zhang, Y., and Che, H.: Feedback effects of boundary-layer meteorological factors on cumulative explosive growth of PM2.5 during winter heavy pollution episodes in Beijing from 2013 to 2016, Atmos. Chem. Phys., 18, 247–258, https://doi.org/10.5194/acp-18-247-2018, 2018.

5. Li, Z., Guo, J., Ding, A., Liao, H., Liu, J., Sun, Y., Wang, T., Xue, H., Zhang, H., and Zhu, B.: Aerosol and boundary-layer interactions and impact on air quality, Natl. Sci. Rev., 4, 810–833, https://doi.org/10.1093/nsr/nwx117, 2017.

6. Li, T., Wang, H., Zhao, T., Xue, M., Wang, Y., Che, H., and Jiang, C.: The Impacts of Different PBL Schemes on the Simulation of PM2.5 during Severe Haze Episodes in the Jing-Jin-Ji Region and Its Surroundings in China, Adu. Meteorol., http://dx.doi.org/10.1155/2016/6295878, 2016a.

7. Vautard, R., Moran, M. D., Solazzo, E., Gilliam, R. C., Matthias, V., Bianconi, R., Chemel, C., Ferreira, J., Geyer, B., Hansen, A. B., Jericevic, A., Prank, M., Segers, A., Silver, J. D., Werhahn, J., Eolke, R., Rao, S. T., and Galmarini, S.: Evaluation of the meteorological forcing used for the Air Quality Model Evaluation International Initiative (AQMEII) air quality simulations, Atmos. Environ., 53, 15-37, https://doi.org/10.1016/j.atmosenv.2011.10.065, 2012.

**Comment 3:** *In the third conclusion (Line 342-343): The authors argued that "During the heavy pollution process, the calculated momentum and sensible heat fluxes by the Li scheme were better than*

*those by the MM5 scheme generally". If the authors only compared simulated momentum and sensible heat to the observation, why this work emphasized the "heavily polluted conditions"? Future work may consider coupling SL scheme with atmospheric chemistry models to compare the modeled pollutant concentration with observation directly.*

**Response:**

The statement "During the heavy pollution process, the calculated momentum and sensible heat fluxes by the Li scheme were better than those by the MM5 scheme generally" was inaccurate. In fact, the surface turbulent flux affects the stability of atmospheric stratification directly, which further influences the air pollution. The little turbulence flux transfer corresponds to stable atmospheric stratification and which may lead to the heavy pollution. In order to make our meaning clearly, we have rewritten this part in Line 377-384, Page 13 in the revised paper.

Thanks for the referee's kind advice. We are online coupling the new scheme into atmosphere chemical models to compare the modeled pollutant concentration with observation directly and the related results will be discussed in next paper.

**Minor comments:**

**Comment 1:** *Line 65-66: Why is the pollution episode important? The author may need to specify and add more discussion instead of arguing "few studies discussed it based on a pollution episode corresponding various atmospheric states".*

**Response:**

Yes, this part was not clearly descripted. We read some new references (list in the following) and add the related content to explain why the pollution episode is important in Line 76-83, Page 3, instead of "few studies discussed it based on a pollution episode corresponding to various atmospheric states".

References:

1. Wang, H., Tan, S. C., Wang, Y., Jiang, C., Shi, G., Zhang, M., and Che, H. Z.: A multisource observation study of the severe prolonged regional haze episode over eastern China in January 2013, Atmos. Environ., 89, 807-815, https://doi.org/10.1016/j.atmosenv.2014.03.004, 2014.

2. Zhang, B., Wang, Y., and Hao, J.: Simulating aerosol-radiationcloud feedbacks on meteorology and air quality over eastern China under severe haze conditionsin winter, Atmos. Chem. Phys., 15, 2387–2404, http://doi.org/10.5194/acp-15-2387-2015, 2015.

3. Li, T., Wang, H., Zhao, T., Xue, M., Wang, Y., Che, H., and Jiang, C.: The Impacts of Different PBL Schemes on the Simulation of PM2.5 during Severe Haze Episodes in the Jing-Jin-Ji Region and Its Surroundings in China, Adu. Meteorol., http://dx.doi.org/10.1155/2016/6295878, 2016a.

4. Liu, T. T., Gong, S. L., He, J. J., Yu, M., Wang, Q. F., Li, H. R., Liu, W., Zhang, J., Li, L., Wang, X. G., Li, S. L., Lu, Y. L., Du, H. T., Wang, Y. Q., Zhou, C. H., Liu, H. L. and Zhao, Q. C.: Attributions of meteorological and emission factors to the 2015 winter severe haze pollution episodes in China's Jing-Jin-Ji area, Atmos. Chem. Phys., 17, 2971–2980, https://doi.org/10.5194/acp-17-2971-2017, 2017.

**Comment 2:** *Line 172-180: The observation and method should be introduced in further details. What is the spatial representativeness of the station? Can it represent the whole east China? If not, should add more cases in other parts of China or considering changing this word. What is the measuring height for the fluxes? (Could refer to Liu et al. 2016 as an example for the introduction)*

**Response:**

This suggestion is very valuable and we revised the manuscript as following according to this

suggestion and the recommended reference.

We have added some introduce about the observation and method in details. Please see Line 183-202, Page 7. The measuring height for the fluxes in Gucheng station is 4 m, which is added in Line 188, Page 7.

Gucheng station is a farmland site where rice is planted in summer and wheat in winter, its surroundings are mainly farmland and scattered villages which represents suburban with smooth surface and it does not represent the whole east China. In fact, our study focuses on "Jing-Jin-Ji" region in east China. We changed "east China" as "Jing-Jin-Ji" in the manuscript; According to the referee's comment, the similar experiment and discussion at Beijing station which represents megacity with rough surface, were added in the revised manuscript (Fig. 9), and the difference of the two schemes in Jing-Jin-Ji region (Fig. 10) was also added in the manuscript.

**Comment 3:** *Line 182-189: The data processing should be explained in further details and add more reference in data processing methods (Line 182-Line 189). For example, how was the quality control conducted? The reference for quality control may be included if they have been applied in the study (e.g., frequency response correction (Moore, 1986) and WPL correction (Webb et al., 1980), or quality control (Foken et al., 2004)).*
**Response:**

Thanks very much for the references recommended by the referee. We have read these references and explained the data processing in more details (Line 196-202, Page 7) and added the relevant reference in Line 197, Page 7.

**Comment 4:** *Please explain why z = 10 m has been used (line 218)?*
**Response:**

"Considering the lowest level in mesoscale models is usually about 10m, $z = 10$m is set as the reference height." The revised part can be found in Line 244, Page 9.

**Comment 5:** *What variables have been used in Li and MM5 schemes? In the third part (Observational data and methods), the paper only introduced the data acquired from the Gucheng station, without specifying what variables would be used in the two schemes.*
**Response:**

Both Li and MM5 schemes use same variables acquired from Gucheng and other stations. The variables used in the two schemes were add in the paper "The measured meteorological variables including wind speed and direction, temperature, humidity, pressure, radiation are used to calculate the momentum and sensible heat fluxes both in the Li and MM5 schemes." The new revision can be seen in Line 189-191, Page7.

**Comment 6:** *Straight from 5. Line 247, the authors mentioned: "Given the observational data, a dataset of Z0m (Z0h) then is generated". What variables were used in calculating Z0m and Z0h? This may be clarified in the third part (observational data and methods).*
**Response:**

The specific variables are added including pressure, temperature, humidity, wind speed and direction, flux for momentum and sensible heat at 4m height, surface skin temperature and we moved this part to the Section 3.3 (Determination of roughness length $z_{0m}$ $(z_{0h})$ ) according to the referee's

suggestion. The revised details can be found in Line 214-223,Page 8.

**Comment 7:** *Line 250 to Lint 264: The author may consider comparing their conclusion with analysis from other papers (Chen et al. 2009; Chen et al. 2011). The reference used here is somewhat out of date.*

**Response:**

This part (Section 4.3) mainly compared the Li and MM5 schemes in flux calculation during observation. We have not any references in this section, so we are not sure which reference used here is somewhat out of date. However, we read the two papers and added the two references in our manuscript (Line 282-283, Page 10) for the related content with our study.

**Comment 8:** *In the Fig. 4, the authors showed the effect of the roughness length on flux calculation by choosing different z0m values. Since the z0m and Z0h has already been determined in the crop field, I feel it may not be necessary to discuss the influence of roughness length on the calculation of turbulent flux.*

**Response:**

$z_{0m}$ is mainly determined by land-cover type and canopy height, but $z_{0h}$ is also affected by nature of the atmospheric flow (Brutsaert, 1975), the underlying surface is neither the only one, nor the most important factor for $z_{0h}$. Furthermore, the different treatment of $z_{0m}$ and $z_{0h}$ in different schemes (e.g., Li and MM5) has great impact on flux calculation and this is also the main reason why the Li scheme is superior to MM5 discussed in the manuscript (Figs. 5, 7, and 8). Therefore, it is necessary and important to discuss the effects of $z_{0m}$ and $z_{0h}$ on the calculation of turbulent flux.

Reference:

Brutsaert, W., The roughness length for water vapor, sensible heat, and other scalars, J. Atmos. Sci., 32, 2028 – 2031, 1975.

**Comment 9:** *Line 315-316: In the previous results and discussion, the authors only analyzed the superiority of Li scheme in modeling sensible heat and momentum flux. More analysis is needed discussing the SL flux influence the air pollution process should be illustrated before concluding "the superiority of Li scheme in the air pollution modeling."*

**Response:**

The expression of the paragraph "Therefore, the superiority of the Li scheme in the air pollution process, especially in this stage is of great reference value for improving the forecast of pollutant concentration in the current air quality model. In stage 3, the difference between the two schemes is not obvious" is not clear enough. Offline study of the two schemes in this work could not draw the conclusion "the superiority of Li scheme in the air pollution modeling", but it is expected to better performance in online simulation of $PM_{2.5}$ based on its obvious superiority in the offline study results. So, this paragraph was replaced by "The error of Li is much less than that of MM5. Considering the importance of atmospheric stratification in the generation and accumulation of $PM_{2.5}$ in stage 2, the Li scheme is expected to show better performance in online simulation of $PM_{2.5}$ than MM5." The details can be found in Line 330-332, Page 12 in the revised paper.

---

## Author Response (AR1)

Reply to Anonymous Referee #3:

We sincerely appreciate for the reviewer's careful dealing of our manuscript and valuable comments. We have read and discussed these comments in detail and answer them one by one in the followings. The corresponding revisions have also been added in the manuscript.

**General comments by Referee #3**

*This study evaluated two surface layer schemes offline, and showed that the new Li scheme presents a better performance over the classic MM5 scheme in terms of the momentum and sensible heat fluxes. Given the importance of the surface exchange processes in a pollution episode and pollution forecast, an accurate representation of the surface processes would be required in a numerical model. This manuscript gave a rather good description about the two schemes, and the results did show that Li scheme may produce better agreement with observations especially in the transition stage of a haze episode. However, I have a few major concerns about this paper:*

**Comment 1:** *What is the scientific contribution of this paper? The authors have well-addressed my comment in the quick report about the new improved surface layer scheme. However, as a scientific paper, I think the authors should also discuss and summarize the scientific findings of this study besides discussing the performance of the two schemes. For example,*

**Response:**

Thanks for the referee's advice. We have added some relevant content to strengthen the scientific contribution of our paper, and rewritten the conclusion and abstract of the manuscript. The scientific findings of this study are: (1) $z_{0m}$ and $z_{0h}$ have important effects on turbulent flux calculation in the SL schemes and ignoring the difference between $z_{0m}$ and $z_{0h}$ in the MM5 scheme could lead to large errors in calculation of sensible heat fluxes. In addition, ignoring the effect of the RSL in schemes may also results in certain bias of momentum and sensible heat fluxes in megacity regions which represent the rough underlying surface; (2) the magnitude of roughness lengths has significant influence on the two schemes. The difference of momentum and sensible heat fluxes calculated by Li and MM5 was much bigger over rough surface than over smooth surface, which suggests that the MM5 scheme probably induces bigger error in megacities with rough underlying surface than it in suburban area with smooth surface; (3) Li scheme better characterized the evolution of atmospheric stratification which is closely related to the haze pollution, compared with the MM5 scheme. This advantage was the most prominent in the transition stage from unstable to stable atmospheric stratification corresponding to the PM$_{2.5}$ accumulation. The offline study of the two SL schemes in this paper showed the superiority of Li scheme for surface flux calculation corresponding to the PM$_{2.5}$ evolution during the haze episode in Jing-Jin-Ji in east China. The study results offer the prerequisite and a possible way to improve PBL diffusion simulation and then PM$_{2.5}$ prediction, which will be achieved in the follow-up work of online integrating of the Li scheme into the atmosphere chemical model.

**1)** *How does the roughness length affect the turbulent fluxes and hence the pollution?*

**Response:**

The surface parameters roughness lengths ($z_{0m}$ and $z_{0h}$) directly affect the calculation of both the surface layer scheme and the turbulent flux (momentum flux and sensible heat flux) which control the atmospheric stratification closely related to the haze pollution. To be specific, ignoring the difference between $z_{0m}$ and $z_{0h}$ in the MM5 scheme induced an obvious overestimation in calculating sensible heat flux (Fig. 6b). Instead, reasonable values of $z_{0m}$ and $z_{0h}$ in the Li scheme produced better agreement with observations (Figs. 6a-b). Furthermore, the Li scheme better characterized the evolution of atmospheric stratification from unstable to stable condition (Figs. 7-8), due to the reasonable treatment of the two parameters.

In addition, we added some new content to further discuss the important role of the roughness lengths (Figs. 9). The result showed that the differences of momentum and sensible heat fluxes calculated by Li and MM5 were much bigger in Beijing than that in Gucheng. This suggests that the MM5 scheme probably induces bigger error in megacities with rough surface (e.g., Beijing) than it in suburban area with smooth surface (e.g., Gucheng) due to the irrational algorithm of the MM5 scheme itself and the ignoring difference between $z_{0m}$ and $z_{0h}$.

The study results above indicate the important role of the roughness lengths in turbulent fluxes and also suggest the improving possibility of severe haze prediction in Jing-Jin-Ji in east China by coupling the Li scheme with more reasonable treatment of roughness lengths and algorithms into the atmosphere chemical model online.

**2)** *Does the roughness length plays a more important role in the transition stage of a pollution episode? And why?*
**Response:**

Yes. The Li scheme performed the best in the transition stage of the pollution episode at Gucheng station, compared with the MM5 scheme, and the biggest difference between Li and MM5 is the treatment of roughness lengths. Therefore, it can be inferred that the roughness lengths play a more important role in the transition stage of the pollution episode at Gucheng station. The results of Jing-Jin-Ji region were similar with Gucheng (Fig. 10 added in the revised manuscript).

In addition, we have added some new experiments to illustrate the important role of this surface parameter (Figs. 4-5, which were revised and add the contrast experiments of RSL). The results showed that the roughness lengths have a much higher effect on the momentum and sensible heat transfer than other factors such as the RSL as well as the universal function. We expect to find more observations to further evaluate it.

**Comment 2:** *There are a lot of grammar mistakes. Please carefully edit the manuscript to improve the language to ensure a better delivery of the scientific ideas and findings to the audience.*
**Response:**

We are so sorry for that. We have a careful examination of the full text including the tables and figures and revised the manuscript to ensure a better delivery of the scientific ideas and findings to the audience. All the changes can be seen in the manuscript with marked-up version.

Reply to Anonymous Referee #2:

We sincerely appreciate for the reviewer's careful dealing of our manuscript and valuable comments. We have read and discussed these comments in detail and answer them one by one in the followings. The corresponding revisions have also been added in the manuscript.

**General comments by Referee #2**

*This work evaluated the performance of a new surface layer scheme (Li) and a widely applied scheme (MM5) in simulating the momentum and sensible heat fluxes. Using the observational data in Gucheng station located in the southwest of Beijing from Dec 1, 2016, to Jan. 9, 2017, The authors found the Li scheme generally performed better than MM5 in calculating SL fluxes during the heavy pollution process. The study fits within the scope of the journal, and the manuscript is generally well written. The result presented is interesting as it shows the SL scheme performance in a polluted case. However, I found that some key details on the introductions are lacking and some of the discussions are not very well grounded.*

**Response:**

Thanks for the affirmation to our work. Yes, we agreed that some key points on the introduction were not enough and some discussions were not very well grounded. We have examined the introduction as well as whole text and the corresponding revisions have been added in the manuscript.

**Comment 1:** *The author should explicitly explain the scientific meaning of the paper. Since Li scheme has been published and evaluated in Li et al. (2014; 2015), why do we need additional evaluation using the observation during a severe haze episode from Gucheng station? I believe this evaluation may be necessary, but the authors need to illustrate clearly the specialty of this case. Also, the word "east China" appears several times in the paper. How did the author conclude Li generally performed better than MM5 in winter in east China since they only did one case in Beijing?*

**Response:**

The Li scheme consists of two parts (Li et al., 2014; 2015). The first part (Li et al., 2014) focused on the stable stratification, while the latter (Li et al., 2015) focused on the unstable conditions. The two parts have not been consolidated into a complete scheme in previous studies. In our study, the two parts were consolidated into one for both stable and unstable conditions. Furthermore, previous work (Li et al., 2014; 2015) was only compared with other iterative or non-iterative schemes. They have neither been compared with actual observations, nor evaluated under the transition process from unstable to stable conditions, which is essential and meaningful. We didn't introduce clearly in our old manuscript and we re-summarized this content in Line 74-83, Page 3 in the revised manuscript.

Yes, the word "east China" is not accurate in this paper. In fact, our study focuses on the Jing-Jin-Ji region in east China. We have replaced "east China" with "Jing-Jin-Ji" in the whole manuscript; In addition, we added Beijing station as well as Jing-Jin-Ji region to discuss the performance of Li and MM5 schemes for different land-cover types (added Figs. 9-10 and the related contents in the revised manuscript).

**Response:**

Thanks very much for the references recommended by the referee. We have read these references and explained the data processing in more details (Line 196-202, Page 7) and added the relevant reference in Line 197, Page 7.

**Comment 4:** *Please explain why z = 10 m has been used (line 218)?*

**Response:**

"Considering the lowest level in mesoscale models is usually about 10m, $z = 10\text{m}$ is set as the reference height." The revised part can be found in Line 244, Page 9.

**Comment 5:** *What variables have been used in Li and MM5 schemes? In the third part (Observational data and methods), the paper only introduced the data acquired from the Gucheng station, without specifying what variables would be used in the two schemes.*

**Response:**

Both Li and MM5 schemes use same variables acquired from Gucheng and other stations. The variables used in the two schemes were add in the paper "The measured meteorological variables including wind speed and direction, temperature, humidity, pressure, radiation are used to calculate the momentum and sensible heat fluxes both in the Li and MM5 schemes." The new revision can be seen in Line 189-191, Page7.

**Comment 6:** *Straight from 5. Line 247, the authors mentioned: "Given the observational data, a dataset of Z0m (Z0h) then is generated". What variables were used in calculating Z0m and Z0h? This may be clarified in the third part (observational data and methods).*
**Response:**

The specific variables are added including pressure, temperature, humidity, wind speed and direction, flux for momentum and sensible heat at 4m height, surface skin temperature and we moved this part to the Section 3.3 (Determination of roughness length $z_{0m}$ ($z_{0h}$) ) according to the referee's suggestion. The revised details can be found in Line 214-223,Page 8.

**Comment 7:** *Line 250 to Lint 264: The author may consider comparing their conclusion with analysis from other papers (Chen et al. 2009; Chen et al. 2011). The reference used here is somewhat out of date.*
**Response:**

This part (Section 4.3) mainly compared the Li and MM5 schemes in flux calculation during observation. We have not any references in this section, so we are not sure which reference used here is somewhat out of date. However, we read the two papers and added the two references in our manuscript (Line 282-283, Page 10) for the related content with our study.

**Comment 8:** *In the Fig. 4, the authors showed the effect of the roughness length on flux calculation by choosing different z0m values. Since the z0m and Z0h has already been determined in the crop field, I feel it may not be necessary to discuss the influence of roughness length on the calculation of turbulent flux.*
**Response:**

$z_{0m}$ is mainly determined by land-cover type and canopy height, but $z_{0h}$ is also affected by nature of the atmospheric flow (Brutsaert, 1975), the underlying surface is neither the only one, nor the most important factor for $z_{0h}$. Furthermore, the different treatment of $z_{0m}$ and $z_{0h}$ in different schemes (e.g., Li and MM5) has great impact on flux calculation and this is also the main reason why the Li scheme is superior to MM5 discussed in the manuscript (Figs. 5, 7, and 8). Therefore, it is necessary and important to discuss the effects of $z_{0m}$ and $z_{0h}$ on the calculation of turbulent flux.
Reference: Brutsaert, W., The roughness length for water vapor, sensible heat, and other scalars, J. Atmos. Sci., 32, 2028 – 2031, 1975.

**Comment 9:** *Line 315-316: In the previous results and discussion, the authors only analyzed the superiority of Li scheme in modeling sensible heat and momentum flux. More analysis is needed discussing the SL flux influence the air pollution process should be illustrated before concluding "the superiority of Li scheme in the air pollution modeling."*
**Response:**

The expression of the paragraph "Therefore, the superiority of the Li scheme in the air pollution process, especially in this stage is of great reference value for improving the forecast of pollutant concentration in the current air quality model. In stage 3, the difference between the two schemes is not obvious" is not clear enough. Offline study of the two schemes in this work could not draw the conclusion "the superiority of Li scheme in the air pollution modeling", but it is expected to better performance in online simulation of PM$_{2.5}$ based on its obvious superiority in the offline study results. So, this paragraph was replaced by "The error of Li is much less than that of MM5. Considering the importance of atmospheric stratification in the generation and accumulation of PM$_{2.5}$ in stage 2, the Li scheme is expected to show better performance in online simulation of PM$_{2.5}$ than MM5." The details can be found in Line 330-332, Page 12 in the revised paper.
Please note that all revised manuscript mentioned above is the final clean manuscript version.

[revised manuscript text omitted]

In many numerical models, surface momentum, heat and moisture fluxes calculated by a SL scheme are coupled to a Land Surface Module, which in turn provides input to the PBL module. Therefore, an adequate SL scheme is crucial for the model performance (Jiménez et al., 2012). It was reported that the difference of 2 m temperature modeling in three PBL schemes is due to different calculation of sensible heat fluxes in the SL (Hu et al., 2010). Tymvios et al.(2017) evaluated the perfomence of Weather Research and Forecasting (WRF) model with a combination of several PBL and compatible SL schemes and emphasized the importance of SL schemes.

The bulk aerodynamic formulationMost SL schemes used in numerical models are bulk algorithms which are based on Monin-Obukhov similarity theory (hereinafter MOST, Monin and Obukhov, 1954) is usually employed to calculate surface fluxes in numerical models. In a bulk algorithm, vertical fluxes in the SL can be considered constant. The effects of shear stress and buoyancy on turbulent transport are discussed with the method of similarity theory and dimensional analysis. Turbulent fluxes in models are parameterized by wind, temperature, humiditymoisture in the lowest layer in model and temperature and humidity in surface., surface skin temperature and humidity. Many international scholars verified the MOST using of field experiments and then proposed the universal functions, the commonly used of which is Businger-Dyer (BD) equation (Businger, 1966; Dyer, 1967). With the development of observation technology, the coefficients in the BD equation have been further modified (e.g.,Paulson, 1970; Webb, 1970; Businger et al., 1971; Dyer, 1974; Högström, −1996). In addition to the BD equation, some other schemes have been put forward and they may performed better especially for the strongly stable stratification (e.g.,Holtslag and De Bruin, 1988;, Beljaars and Holtslag, 1991;, Chenge and Brutsaert, 2005). The schemes can be divided into two types according to the computing characteristics. One type is called as iterative algorithm (e.g.,Paulson, 1970; Businger et al., 1971; Dyer, 1974; Högström, 1996; Beljaars and Holtslag, 1991), and it keeps the MOST completely with less approximation so that the results can be more precise. However, it needs to take much more steps to converge and hence the CPU time is consuming which reduces the computationalaffects the ability and efficiency of modeling (Louis, 1979; Li et al., 2014); The other one is called as non-iterative algorithm (e.g.,Louis et al., 1982; Launiainen, 1995; Wang et al., 2002; Wouters et al., 2012). Due to the approximate treatment, tThere is no need for loop iteration in the calculation due to the approximate treatment. ItThis algorithm is much simpler and less CPU time-consuming, but the results are based on the loss of the calculation accuracy.it may lead to a lower accuracy of the results.

Although many researches above focused on the effects of the SL schemes on PBL and meteorological elements, few studies discussed it based on a pollution episode corresponding various atmospheric states. The turbulent exchange of momentum, heat, and moisture at the ground surface is more important than large scale transport for the accumulation and

[revised manuscript text omitted]
 9, 2017. ~~(GC), which is in China Atmosphere Watch Network (CAWNET) and located in the southwest of Beijing about 110km, at 115.40° E, 39.08° N. In winter, the station surface was covered with wheat and the surrounding areas were mainly farmland and scattered villages (Fig. 1). The eddy correlation flux measurement system is mainly composed of a three-dimensional (3D) Temperature measurement with a sonic anemometer (CSAT3) and a fast response infrared gas~~

analyzer (LI-7500) at 4m height. The data was collected from December 1, 2016 to January 9, 2017 including momentum fluxes, heat fluxes, wind speed and wind direction, air temperature, density of air and vapor, pressure with 30 minutes interval. Besides, there were radiation data provided by the net radiation sensor (CNR1) including the surface upward long wave radiation and the long wave radiation incident to the ground surface and $PM_{2.5}$ data provided by the Environmental Protection Station of China's Ministry of Environmental Protection (EPS/CMEP). Gucheng station (115.40 °E, 39.08 °N) is located at Gucheng County, Baoding, Hebei province and it is about 110km southwest of Beijing (Fig. 1a). This station has a farmland site where rice is planted in summer and wheat in winter. The surroundings are mainly farmland and scattered villages (Fig. 1b). At Gucheng station, the momentum and sensible heat fluxes near surface were measured by the eddy correlation flux measurement system. The system is mainly composed of a sonic anemometer (CSAT3) and a gas analyzer (LI-7500). They are set up at 4m height above surface ground. The measured fluxes are used to evaluate the two schemes as well as estimate the roughness lengths. The measured meteorological variables including wind speed and direction, temperature, humidity, pressure, radiation are used to calculate the momentum and sensible heat fluxes both in the Li and MM5 schemes. Note the observed meteorological data were from Gucheng station and national basic automatic weather stations in Jing-Jin-Ji in east China, respectively. Hourly surface $PM_{2.5}$ mass concentration in Baoding and Beijing from China National Environmental Monitoring Centre (http://www.cnemc.cn/) were also used in this paper.

**3.1 Data processing**

In order tTo obtain accurate flux data, it needs quality control has been performed forof the observational data, including: (1) eliminated the outliers and the data in rainy days: (2) double rotation and WPL correction (Webb et al., 1980); (3) omit the dataset when the wind speed are less than 0.5m/s., as well as correcting momentum by using a double axis rotation for the sonic anemometer tilt correction and correcting sensible heat fluxes by modifying sonic virtual temperature. In addition, the wind field especially the wind direction has a great impact on the value of $z_{0m}$, so it is necessary to understand the situation at Gucheng station. we considered the effect of wind field on the roughness length. Fig. 2 shows the distribution frequency of wind speed and wind direction at GCGucheng during the observations (December 1, 2016 ~ January 9, 2017). The wind speed is stable during this period and the maximum is no more than 5 m/s and most of them are about 1 ~ 2 m/ s. The wind direction is relatively uniform except for the southeast wind (135 ° degrees). Therefore, to avoid the measurement error of the instrument, the wind speed data less than 0.5m/s are eliminated.

**3.2 Determination of surface skin temperature**

The surface skin temperature at Gucheng station error caused by the CSAT3 is too large to be taken to calculate the flux as input. Therefore, the surface skin temperature 
[revised manuscript text omitted]
. Thus, assume $z_{0m}$ and $z_{0h}$ are two fixed values. Given the observational data, a dataset of $z_{0m}$ ($z_{0h}$) then is generated. Finally take median of the dataset as typical values of $z_{0m}$ and $z_{0h}$ for GC site: $z_{0m} = 0.0419\text{m}$, $z_{0h} = 0.0042\text{m}$. These results are comparable to the typical values for agricultural fields ($z_{0m} = 0.05$, $z_{0m}/z_{0h} = 10$) discussed above. Therefore, the results are considered credible.~~

**4.~3~2 Comparison of  momentum and sensible heat flux**

Using the  _obtained_ roughness length_s_ and the  observations, the _momentum and sensible heat flux were calculated by the_ Li and MM5 schemes_. Firstly, $z_{0m}$ and $z_{0h}$ were set as 0.0419 and 0.0042 respectively in the Li scheme, $z_0$ was equal to $z_{0m}$ in the MM5 scheme to calculate the momentum and sensible heat fluxes and the_ results are shown in Figs. 6a and 6b. _It can be seen that C_ompared with MM5, Li performs better with higher regression coefficient and determination coefficient. For momentum fluxes, the regression coefficient _by_ Li is 0.6795 and that _by_ MM5 is 0.5598, indicating that the error of Li is 12% lower than that of MM5. For sensible heat fluxes, the regression coefficient _by_ Li is 0.7967 and that _by_ MM5 is 1.7994. The latter is much larger than 1_, that is,_  the MM5 scheme _obviously_ overestimate _the sensible heat due to it does not distinguish $z_{0h}$ from $z_{0m}$._ _Then, make $z_0$ equal to 0.0042 in the MM5 scheme to re-calculate the sensible heat fluxes as shown in Fig. 6c. It can be seen the result has a great improvement after modifying $z_0$ value and the regression coefficient by MM5 is 0.7363, indicating that the error was reduced by 54% after considering the $z_{0h}$ effect. The result indicates that $z_{0h}$ plays a key role in both the SL scheme and the sensible heat flux (Chen and Zhang, 2009; Chen et al., 2011)._~~That is due to no distinction of roughness length in the MM5 scheme. In order to compare the difference of two schemes without considering the effect of roughness length, take $z_0 = z_{0h} = 0.0042$ in the MM5 scheme to calculate the sensible heat fluxes as Fig. 6c. Compared with Fig. 6b, there is a great improvement after modifying $z_0$ value that the regression coefficient in MM5 becomes 0.7363, which is indicated that the error of calculated sensible heat flux by MM5 was reduced by 54% after discriminating $z_{0h}$ from $z_{0m}$.in6in(including the selection of universal functions and the consideration of the RSL effect)e~~ _of_ MM5 scheme.

**4.3 The specific performance of the two scheme in the severe haze pollution**

[revised manuscript text omitted]

**5 Conclusions**

Using the observed momentum and sensible heat fluxes, together with conventional meteorological data including pressure, temperature, humidity and wind speed from December 1, 2016 to January 9, 2017, including a severe pollution episode from December 13 to 23, 2016, the differences and the performance of the two surface schemes were discussed and evaluated in this paper. The evolution process of atmospheric stratification from unstable to stable corresponding to $PM_{2.5}$ increasing was mainly discussed. The contributions of roughness lengths ($z_{0m}$ and $z_{0h}$) and other factors in the SL schemes to the momentum and sensible heat flux calculation were also discussed in details. The results are summarized as follows:

1) $z_{0m}$ and $z_{0h}$ have important effects on turbulent flux calculation in the SL schemes. Different values of $z_{0m}$ and $z_{0h}$ in the schemes could induce great changes in flux calculation, indicating that it is very necessary and important to distinguish $z_{0h}$ from $z_{0m}$. Ignoring the $z_{0h}$ effect in the MM5 scheme led to large errors in calculation of sensible heat fluxes and this error in Gucheng is 54%. Besides the roughness lengths, the algorithms of two schemes are also one of important factors. In addition, ignoring the effect of the RSL in schemes may also results in certain bias of momentum and sensible heat fluxes in megacity regions which represent the rough underlying surface. ~~$z_{0m}$ and $\frac{z_{0m}}{z_{0h}}$ both reflect the condition of underlying surface and impact flux calculation greatly. Under the same condition, the larger $z_{0m}$ (indicating rougher surface) is, the larger the calculated fluxes are. The fluxes over large cities ($z_{0m}=1$) is quite different from those over agricultural fields ($z_{0m}=0.05$, similar to the value at GC). When $z_{0m}$ is larger, the value of $\frac{z_{0m}}{z_{0h}}$ should be larger, and the larger the value of $\frac{z_{0m}}{z_{0h}}$ is, the greater the differences of calculated fluxes are. Especially, for a super city like Beijing, the value of $\frac{z_{0m}}{z_{0h}}$ may be much larger than $10^6$ and ignoring the difference between z0m and z0h may lead to much uncertainties in flux calculation. It is very necessary to distinguish between $z_{0m}$ and $z_{0h}$ in SL scheme, which is probably beneficial to improve simulation of regional atmosphere stratification over urban agglomeration with rough surface and then $PM_{2.5}$ during hazes.~~

2) The effect of $z_{0m}/z_{0h}$ on turbulent fluxes is closely related to the land-cover types ($z_{0m}$). A rough land-cover type (large $z_{0m}$) should be accompanied by a large value of $z_{0m}/z_{0h}$. The differences of momentum and sensible heat fluxes calculated by Li and MM5 were much bigger in Beijing than that in Gucheng. This suggests that the MM5 scheme probably induces bigger error in megacities with rough surface (e.g., Beijing) than it in suburban area with smooth surface (e.g., Gucheng) due to the irrational algorithm of MM5 scheme itself and the ignoring difference between $z_{0m}$ and $z_{0h}$.

~~2) It could be seen from the regression coefficients and determination coefficients between calculated fluxes by the two schemes and observed fluxes of 40 days that the Li scheme was better than the MM5 scheme in general. For the momentum fluxes, the determination coefficients of Li and MM5 was about 0.41 and 0.40. Both schemes passed the significance level of 99.9%. The regression coefficient of Li was 0.68, and it generally reduced the error by 12% compared with MM5. When $z_{0m}$ and $z_{0h}$ took the same value ($z_0 = z_{0m} = 0.0419$) in MM5, the sensible heat fluxes were obvious overestimated. When $z_{0h}$ was taken into account ($z_0 = z_{0h} = 0.0042$) in MM5, the calculated fluxes were significant improved and the error was reduced by 54%. However, this error was still higher about 5% compared with the Li scheme, illustrating that apart from the impact of roughness length, the different algorithms of the two schemes also achieves obvious differences in calculated fluxes.~~

3) The Li scheme generally performed better than the MM5 scheme in the calculation of both the momentum flux and the sensible heat flux compared with observations at Gucheng station. The Li scheme made a better description in atmospheric stratification which is closely related to the haze pollution, compared with the MM5 scheme. This advantage of Li scheme was the most prominent in the transition stage from unstable to stable atmospheric stratification corresponding to the $PM_{2.5}$ accumulation. In this stage, the momentum flux calculated by Li was overestimated by 7.68% and this overestimation by MM5 was up to 45.56%; the sensible heat flux by Li was underestimated by 33.84% while this underestimation by MM5 was even up to 76.88%. In most Jing-Jin-Ji region, the momentum fluxes calculated by Li were less than that by MM5 and the sensible heat fluxes by Li were larger than that by MM5, which was consistent with Gucheng.

~~3) During the heavy pollution process, the calculated momentum and sensible heat fluxes by the Li scheme were better than those by the MM5 scheme generally. Especially in the PM2.5 accumulated stage, the advantages of Li were more prominent. Compared with MM5, the probability distributions of both the momentum and sensible heat flux bias of Li tended to cluster in a narrower range centered by 0. The calculated momentum fluxes by Li were overestimated by 7.68% and this overestimation by MM5 was up to 45.56%. The calculated sensible heat fluxes by Li were underestimated by 33.84% while this underestimation by MM5 was even up to 76.88%.~~

The offline study of two SL  scheme in this paper showed the superiority of the Li  scheme for surface flux calculation corresponding to the $PM_{2.5}$ evolution during the haze episode in Jing-Jin-Ji in east China.

from unstable to stable stratification. However, the comparison of the two schemes focusing on more underlying surfaces (e.g., super cities and agricultural fields) could not be conducted at present due to the shortage of observed fluxes data, which should be discussed in detail in next paper when the sufficient data is available. The offlinestudy results of this paper only offer prerequisite a basic and a possible way to improve PBL diffusion simulation and then PM$_{2.5}$ prediction, which will be achieved in the follow-up work of online integrating of the Li scheme into the atmosphere chemical model.

**Acknowledgments**

The study was supported by the National Key Project (2016YFC0203306)National Key Project of HePAP (JFYS2016ZY01002213), the National (Key) Basic Research and Development (973) Program of China (2014CB441201), the National Key R & D Program Pilot Projects of China (2016YFC0203304)

[revised manuscript text omitted]

---

## Author Response (AR2)

We sincerely thank the reviewers for the time and valuable comments, which have led us to a substantially improved version of the paper. We carefully considered all of these comments and revised the manuscript thoroughly. Our detailed responses to the referee's questions and comments are listed below, and the modification has been marked in the following manuscript.

**Reply to Anonymous Referee #2:**

The authors addressed all my issues in the response. The scientific contribution has been discussed and emphasized in the revision. The manuscript may be published after the following minor changes:

**Comment 1.** Line 306: "Fig. 8 shows …" The Fig. 8 should be Figure. 8 when it comes at the beginning of a sentence. Please also revise other abbreviation according to ACP manuscript preparation guidelines.

**Response:**

Thanks for pointing this out. We have revised this nonstandard writing as well as other abbreviation in the full text, according to the referee's advice and ACP manuscript preparation guidelines.

**Comment 2.** Line 311: "The probabilities of bias by Li and MM5 within ±2.5W·m-2 are 32.54% and 13.49%, respectively." Spaces should be included between number and unit.

**Response:**

We have corrected this mistake as well as the similar problems in the full text according to the referee's kind advice.

**Comment 3.** Figures: Please clear the right boundary of Figure 9.

**Response:**

We have cleared the right boundary of Figure 9 in the revised manuscript.

**Reply to Anonymous Referee #3:**

The authors were very responsive and have done a lot of work to address the concerns from the reviewers. All my major questions and concerns have been well addressed in details. The paper has been significantly improved and I recommend acceptance for publication after some minor/technical revisions/corrections. Note that there are still a lot of grammar errors or typos and I may not be able to identify all of them. A thorough editing is still required.

Minor suggestions or technical corrections:

**Comment 1.** Abstract: Since the title mentions about the severe haze pollution, it should be identified in the abstract that the evaluation was done for haze cases.

**Response:**

Thanks for the referee's advice. We have changed the third sentence to "The differences of two surface layer schemes (the Li and MM5 scheme) were discussed and the performance of the two schemes focusing on a heavy haze episode was mainly evaluated based on the observed momentum and sensible heat fluxes in Jing-Jin-Ji in east China." in Lines 21-23.

**Comment 2.** Lines 19-21, "The pollutants prediction by atmosphere chemical model exist obvious deficiencies, which may be closely related to the uncertainties of the momentum and sensible heat fluxes calculated in the surface layer": Here "prediction", not "pollutants", is a singular subject and hence a singular verb "exists" should be used.

**Response:**

The word "exist" has been changed to "exists" in Line 20.

**Comment 3.** Line 26, "the algorithms of universal functions": universal functions for what? Surface fluxes? Please specify.

**Response:**

Yes. It is universal functions for surface turbulent fluxes. So we have added "for surface turbulent fluxes" after "universal functions" in Line 27.

**Comment 4.** Lines 60-61, "Many international scholars verified the MOST using of field experiments": Do the authors mean "Many international scholars verified the MOST using field experiments"?

**Response:**

Yes. We have removed the "of" in this sentence in Line 62.

**Comment 5.** Line 75, "have" should be "has".

**Response:**

The word "have" has been changed to "has" in Line 76.

**Comment 6.** Lines 80-82, "The observed momentum and sensible heat flux data covering once complete haze process at Gucheng station was used to evalute the two schemes focsuing on the transition stage from unstable to stable atmospheric stratification corresponding to the PM2.5 accumulation": "once" should be "one"; "was": should be "were"; "evaluate" should be "evaluate"; "focsuing" should be "focusing".

**Response:**

We have rewritten the sentence as "The observed momentum and sensible heat flux data covering one complete haze process at Gucheng station were used to evaluate the two schemes focusing on the transition stage from unstable to stable atmospheric stratification corresponding to the PM2.5 accumulation" in Line 81-83.

**Comment 7.** Lines 90-91: The period "." after the equation should be a comma ",", and "Where" should not be capitalized. Check all the other equations in which "where" is used.

**Response:**

We revised this mistake in Line 92-93 as well as other places where the mistake existed according to the referee's kind advice.

**Comment 8.** Line 92: A comma "," needs to be added in front of "respectively".

**Response:**

We added the "," in front of "respectively" in Line 94.

**Comment 9.** Line 93: The subject "they" can be omitted.

**Response:**

The word "They" has been omitted in Line 95.

**Comment 10.** Line 99: do the authors mean "from at least two levels"?

**Response:**

Yes. Thanks for pointing this out, and we added "from" before "at least two levels" in Line 101.

**Comment 11.** Line 101: "employ" should be "employs".

**Response:**

We have changed the word "employ" to "employs" in Line 101.

**Comment 12.** Line 112: should "universe" be "universal"?

**Response:**

Yes. This mistake has been corrected in Line 114.

**Comment 13.** Line 183: add "were" before "measured".

**Response:**

Thanks for pointing this out. We have added "were" before "measured" in Line 185.

**Comment 14.** Section 3: It is my understanding that this study focuses on the impact of the surface flux parameterization on air pollution modeling. Therefore, the haze pollution case(s) or episode(s) should also be briefly described in this section.

**Response:**

We appreciate the referee's suggestion. Our study indeed focuses on the impact of the surface flux parameterization on air pollution modeling, but there are also some analyses and evaluation on non-pollution period in Section 4. Therefore, we described the haze pollution case after the whole discussion in Section 4, which can make the organization clearer.

**Comment 15.** Line 370: "results in" should be "result in".

**Response:**

We have revised this mistake in Line 374.

We thoroughly checked the full text according to the referees' suggestions and ACP manuscript preparation guidelines, and all revision details can be seen in the marked-up manuscript version below.

[revised manuscript text omitted]

---

## Author Response (AR3)

**Comments to the Author:**

While the major technical issues have been tackled, there are many English errors or non-standard usages. I just read the title and abstract. A dozen of such problems were noted, as listed below. The paper must be thoroughly edited to drastically improve its English, or it'd be rejected.

**Response:**

We would like to heartily thank the you for your serious review on our work and the valuable comments. We revised the manuscript in accordance with your kind advices and detailed suggestions, and carefully proof-read the manuscript to minimize typographical, grammatical, and bibliographical errors and improve the English in the manuscript. Here below is our description on revision according to your comments. We sincerely hope the correction will meet with approval.

**Comment 1:** The title is misleading, change to "Evaluating the performance of two surface layer schemes for the momentum and heat exchange processes during severe haze pollution in Jing-Jin-Ji in east China"

**Response:**

Thanks for pointing this out, and we have changed the title according to the editor's advice.

**Comment 2:** "Pollutants prediction by atmosphere chemical model exists obvious deficiencies," change to "There have existed some deficiencies in the prediction of pollutants by atmosphere chemical models"

**Response:**

We have revised the sentence in Lines 21-22, and other similar sentences in Lines 51, 56 have also been revised.

**Comment 3:** "The differences of two…" should be "The differences between two"

**Response:**

We changed "of" to "between" in this sentence in Line 24. We also revised similar mistakes in Line 33, Line 170, Line 311, Line 371 and Line 384.

**Comment 4:** "was mainly evaluated 22 based on". To "was evaluated mainly based on …"

**Response:**

Thank you for your advice. We put "mainly" before "evaluated" to illustrate that we evaluated the performances of the two schemes not only for the pollution process but also for the other times. "mainly" is for "a heavy haze episode", not for "the observed momentum and sensible heat fluxes". So we think it would be more appropriate to put "mainly" before "evaluated". We revised this sentence to make the meaning more clear, and the revision is in Lines 24~26.

**Comment 5:** "play a major role in the flux calculation" to " play major roles in the flux calculation"

**Response:**

We changed "play a major role" to "play the major roles" in Line 27. We also corrected other places about singular and plural forms, such as in Line 89, Line 91, Line 272, and Line 335.

**Comment 6:** "Besides the roughness lengths" to "Besides the roughness length"

**Response:**

Thanks for pointing this out and we changed "roughness lengths" to "roughness length" as it is a concept here.

**Comment 7:** "the algorithms of universal functions for" either to " the algorithms for" or " the universal functions for" not both algorithms and functions.

**Response:**

Thanks for the editor's kind advice. We have deleted "of universal functions" in Line 30.

**Comment 8:** "magnitude of $z0\_m$ _and $z0\_h$ _has" to "the magnitudes of $z0\_m$ _and $z0\_h$ _have"

**Response:**

We have revised this mistake in Line 31.

**Comment 9: Change all "compared with" to "comparing with"**

**Response:**

We have changed all "compared with" to "comparing with" in the revised manuscript.

**Comment 10:** "Li scheme better characterized" to "Li scheme is better in characterizing"

**Response:**

We have revised this mistake in Line 34, and other similar parts were also revised.

**Comment 11:** "in the describing of" to either " in the description of" or " in describing .."

**Response:**

Thanks for pointing this mistake out, we have revised all similar problems in whole manuscript.

[revised manuscript text omitted]